# SPARKLING: Balancing Signal Preservation and Symmetry Breaking for Width-Progressive Learning

Qifan Yu [1 2 *]  Xinyu Ma [2]  Zhijian Zhuo [2]  Minrui Wang [2]  Deyi Liu [2]  Shiyi Zhan [2]
Yiyuan Ma [2]  Liang Xiang [2]  Xingyan Bin [2]  Di He [1]

## Abstract

Progressive Learning (PL) reduces pre-training computational overhead by gradually increasing model scale. While prior work has extensively explored depth expansion, width expansion remains significantly understudied, with the few existing methods limited to the early stages of training. However, expanding width during the mid-stage is essential for maximizing computational savings, yet it remains a formidable challenge due to severe training instabilities. Empirically, we show that naive initialization at this stage disrupts activation statistics, triggering loss spikes, while copy-based initialization introduces gradient symmetry that hinders feature diversity. To address these issues, we propose **SPARKLING** (balancing **S**ignal **P**reservation **A**nd symmet**R**y brea**K**ing for width-progressive **L**earn**ING**), a novel framework for mid-stage width expansion. Our method achieves signal preservation via RMS-scale consistency, stabilizing activation statistics during expansion. Symmetry breaking is ensured through asymmetric optimizer state reset and asymmetric learning rate re-warmup. Extensive experiments on dense and Mixture-of-Experts (MoE) models demonstrate that, across multiple width axes and optimizer families, SPARKLING consistently outperforms training from scratch and reduces training cost by up to 35 % under $2\times$ width expansion.

## 1. Introduction

Training Large Language Models (LLMs) remains prohibitively expensive, motivating a growing line of work on

*Work done at ByteDance Seed. [1]State Key Laboratory of General Artificial Intelligence, Peking University [2]ByteDance Seed. Correspondence to: Di He <di_he@pku.edu.cn>, Xingyan Bin <binxingyan@bytedance.com>.

*Proceedings of the 43rd International Conference on Machine Learning*, Seoul, South Korea. PMLR 306, 2026. Copyright 2026 by the author(s).

Progressive Learning (PL) (Kim et al., 2024; Wu et al., 2024; Du et al., 2024), which aims to expand the parameter scale gradually during training instead of training the full scale from scratch (Gong et al., 2019). Existing PL methods have demonstrated notable success in both saving training computation and improving performance, especially through depth-oriented strategies such as layer stacking (Gong et al., 2019; Kim et al., 2024; Du et al., 2024), block insertion (Wu et al., 2024; Yang et al., 2025b), or gradual network growth (Yang et al., 2020).

Width is another crucial dimension for scaling model parameters (Kaplan et al., 2020). Existing studies have made only limited progress in this direction, and a general and systematic mechanism has yet to be established (Chen et al., 2016; 2022; Zhang et al., 2024; Yano et al., 2025; Yao et al., 2024; Yuan et al., 2023). More importantly, previous investigations have been largely limited to expansion during the initial portion of training, e.g., less than 10–30 % of training tokens (Du et al., 2024; Shen et al., 2022). Such early expansion offers negligible computational advantages over training the target-width model from scratch and fundamentally undermines the primary motivation of PL—reducing training costs. To make PL practically viable, width expansion must be conducted during the mid-stage of pre-training. However, it is precisely in this regime that width expansion becomes challenging. We hereby analyze the core challenges as follows.

We identify the first core challenge as *signal preservation* during mid-stage expansion. Prior width-based PL work has largely treated "preservation" as *loss continuity* at the expansion point, i.e., function preservation (FP) where the expanded model is initialized to match the pre-expansion mapping (Chen et al., 2016; 2022; Shen et al., 2022; Wang et al., 2024; Han et al., 2025). While FP is useful, we argue that the core mechanism behind the scenes is whether the expansion preserves the statistical distribution of intermediate activations, most notably the *root-mean-square (RMS)* scale of hidden representations (Zhang & Sennrich, 2019). Concretely, RMS-scale mismatch alters layerwise signal magnitudes and propagates through residual streams; this destabilizes optimization even when no instantaneous loss

spike occurs at the moment of expansion (Bachlechner et al., 2021; Yang et al., 2021).

Based on this perspective, we design RMS-preserving strategies for several standard initialization regimes—copy, random, and zero—ensuring each can be applied without inducing activation-scale shocks. Interestingly, these RMS-preserving variants reveal a counter-intuitive limitation of copying: despite its appeal for function preservation, copy-based expansion can underperform compared to RMS-preserving random or zero initialization in terms of post-expansion recovery. This observation indicates that, beyond maintaining loss and activation scale, some additional and critical issues for copy-based expansion remain unresolved.

This brings us to the second core challenge: copy-based expansion, while strongest in forward continuity, induces *backward symmetry* (Chen et al., 2016). Duplicating channels creates duplicated parameter subspaces that receive identical gradients and thus evolve identically, leaving the new capacity functionally redundant (Wu et al., 2019). Crucially, this symmetry is not an artifact of a specific optimizer: it arises under both element-wise optimizers such as AdamW (Loshchilov & Hutter, 2019) as well as spectral-style updates such as Muon (Jordan et al., 2024; Liu et al., 2025), causing a persistent coupled state in which copied components fail to diversify.

Motivated by these observations, we frame mid-stage width expansion as balancing two complementary principles: *Signal Preservation* and *Symmetry Breaking*. On the preservation side, we enforce RMS-scale consistency during expansion, ensuring that the expanded model maintains stable hidden-state statistics and thereby supports smooth post-expansion optimization. On the symmetry side, we introduce targeted interventions that act only in the backward dynamics while leaving the copied forward function intact: (i) a controlled optimizer state reset for newly introduced parameters to remove inherited symmetric momentum, and (ii) an asymmetric learning rate re-warmup schedule that selectively stimulates the new parameters without globally perturbing the well-adapted pre-trained ones. Together, these mechanisms preserve forward continuity at the moment of expansion, while inducing sufficient asymmetry in subsequent updates for the expanded capacity to diverge and encode meaningful features.

In summary, our contributions can be highlighted as follows:

- We investigate the challenges of width expansion during the critical **mid-stage of pre-training**, a regime largely unexplored in prior work due to stability concerns. We identify that successful expansion hinges on two complementary principles: *Signal Preservation* to stabilize activation statistics, and *Symmetry Breaking* to resolve gradient coupling in copy-based initialization.

- We propose SPARKLING, a practical framework that implements both principles through a suite of concrete mechanisms—including RMS-scale consistency, copy-based initialization, asymmetric optimizer state reset, and asymmetric learning rate re-warmup—that jointly resolve the optimization challenges inherent to expanding deep within the pre-training trajectory.

- We empirically validate the generality of SPARKLING across dense and MoE architectures, multiple width axes (including hidden dimension and MoE expert intermediate dimension), and optimizer families (including AdamW and Muon). Under a fixed token budget, our PL approach consistently **outperforms training the full-scale model from scratch** on downstream evaluations, while **reducing training costs by up to 35 %** when scaling to $2\times$ width, demonstrating both **effectiveness** and **efficiency** of SPARKLING.

## 2. Related Work

Progressive Learning (PL) has emerged as a resource-efficient paradigm that accelerates training by gradually expanding the architecture from a small base model to a target scale during training (Chen et al., 2016; 2022; Gong et al., 2019; Kim et al., 2024). From the perspective of depth expansion, existing strategies typically grow by stacking layers (Gong et al., 2019; Kim et al., 2024; Du et al., 2024) or inserting blocks (Wu et al., 2024; Yang et al., 2025b). Existing approaches for width expansion largely prioritize function preservation (FP) via parameter mapping (Chen et al., 2016), advanced initialization schemes like AKI and its variants (Chen et al., 2022; Zhang et al., 2024; Yano et al., 2025), or temporarily masking new structures (Yao et al., 2024). To address redundancy from simple copying (Chen et al., 2016), various heuristic interventions have been adopted, including uneven splitting (Chen et al., 2016; Wang et al., 2024; Du et al., 2024) and symmetric perturbations (Yuan et al., 2023; Wu et al., 2020; 2019).

Beyond initialization, significant effort has been directed toward stabilizing post-growth optimization dynamics: Wang et al. (2024) advocate for accelerated decay schedules on the premise that expanded models start closer to local optima, while Yuan et al. (2023) utilize weight-norm to rebalance gradient contributions and Shen et al. (2022) propose dynamics-preserving growth operators to align the expanded model's loss trajectory. Other methods attempt to learn growth operators (Wang et al., 2023; Pan et al., 2023) or construct gradient-maximizing weights (Evci et al., 2022).

However, these strategies typically address either forward initialization or backward optimization dynamics in isolation. In this work, we establish a systematic framework balancing both perspectives. We first argue that the mechanism

underlying widely used *function-preserving* initializations is fundamentally *RMS preservation*, and then redesign the optimization procedures to address the symmetry issues that inevitably arise from such preservation-focused initialization strategies.

## 3. RMS Scale Consistency of Activation

### 3.1. Why RMS Mismatch Destabilizes Training

We start by defining the root-mean-square (RMS) magnitude of a vector $\boldsymbol{h} \in \mathbb{R}^d$ as (Zhang & Sennrich, 2019)

$$\text{RMS}(\boldsymbol{h}) := \frac{\|\boldsymbol{h}\|_2}{\sqrt{d}} = \sqrt{\frac{1}{d} \sum_{i=1}^{d} h_i^2}. \qquad (1)$$

Consider a linear layer

$$\boldsymbol{y} = \boldsymbol{W}\boldsymbol{x}, \quad \boldsymbol{W} \in \mathbb{R}^{d_{\text{out}} \times d_{\text{in}}}, \; \boldsymbol{x} \in \mathbb{R}^{d_{\text{in}}}, \; \boldsymbol{y} \in \mathbb{R}^{d_{\text{out}}}. \quad (2)$$

where $\boldsymbol{x}$ and $\boldsymbol{y}$ denote the input and output hidden states, respectively. Our focus is the RMS scale of the activations:

$$r := \frac{s_{\text{out}}}{s_{\text{in}}} = \frac{\text{RMS}(\boldsymbol{y})}{\text{RMS}(\boldsymbol{x})}, \quad r^{(\text{post})} = r^{(\text{pre})}, \qquad (3)$$

requiring this scale to remain unchanged after expansion.

We enforce the RMS invariance in Eq. (3) as a signal preservation constraint. A trained Transformer block implicitly defines an operating regime via its input-output statistics, within which representations are well-formed and features remain meaningful. If width expansion perturbs activation RMS during expansion, post-expansion hidden states can drift away from the pre-expansion scale manifold, causing subsequent blocks to receive out-of-regime inputs. RMS-preserving expansion mitigates this shift by keeping block-wise input/output magnitudes within the original domain, thereby maintaining the fidelity and generalization of the pre-trained function immediately after expansion.

In modern LLMs that adopt pre-normalization (e.g., Qwen3 (Yang et al., 2025a), DeepSeek-V3 (DeepSeek-AI et al., 2025), OLMoE (Muennighoff et al., 2025)), RMS preservation becomes even more critical because the update explicitly couples the residual stream with the branch output. Concretely, in a typical residual block with pre-norm, the hidden state is updated as

$$\boldsymbol{h} \leftarrow \boldsymbol{h} + f(\text{Norm}(\boldsymbol{h})), \qquad (4)$$

where $f(\cdot)$ denotes a residual branch such as an attention or MLP sublayer, and $\text{Norm}(\cdot)$ refers to token-wise feature normalization, i.e., LayerNorm or its variants used in LLMs, most notably RMSNorm (Zhang & Sennrich, 2019).

Here, pre-norm stabilizes the input to $f(\cdot)$ but does not constrain its output scale, so the residual dynamics depend on the ratio $\text{RMS}(f(\text{Norm}(\boldsymbol{h})))/\text{RMS}(\boldsymbol{h})$. After expansion, an RMS mismatch shifts the calibrated mixing between the main path and the transformed branch, making it either overwhelming on the main stream or nearly identity. By preserving RMS through expansion, we keep layerwise dynamics coherent and maintain the balanced residual regime.

The same issue arises in post-normalization variants $\boldsymbol{h} \leftarrow \text{Norm}(\boldsymbol{h} + f(\boldsymbol{h}))$, since the relative weighting inside the residual sum is still governed by the RMS ratio between $\boldsymbol{h}$ and $f(\boldsymbol{h})$. RMS-preserving expansion is therefore architecture-agnostic and remains necessary under post-norm.

### 3.2. RMS-Preserving Expansion

We continue to discuss RMS-preserving width expansion in three cases: (i) *fan-out* expansion, which expands the output dimension ($d_{\text{out}}$), (ii) *fan-in* expansion, which expands the input dimension ($d_{\text{in}}$), and (iii) *RMSNorm weight* expansion, which widens the RMSNorm scale.

In practice, fan-out and fan-in expansions typically appear as a paired transformation across two consecutive layers that share an intermediate width. For example, in an MLP that widens the expert intermediate dimension, the *up* and *gate* projections become fan-out expansions[1], and the subsequent *down* projection becomes fan-in expansion. Similarly, in attention, the *vhead* projection and the output projection likewise form such a paired transformation. Therefore, whenever we widen any width dimension, the two sides are naturally correlated and should be discussed jointly in pairs.

#### 3.2.1. PRELIMINARIES

Under the linear layer defined in Eq. (2), the output activation RMS is given by

$$\text{RMS}(\boldsymbol{y}) = \sqrt{\frac{1}{d_{\text{out}}} \|\boldsymbol{y}\|_2^2} = \sqrt{\frac{1}{d_{\text{out}}} \sum_{i=1}^{d_{\text{out}}} y_i^2}. \qquad (5)$$

Leveraging the property of high-dimensional isotropy in wide neural networks, where feature vectors tend toward asymptotic orthogonality (Bird, 2025; Saxe et al., 2014), we can assume that $\{y_i\}_{i=1}^{d_{\text{out}}}$ are identically distributed and satisfy $\mathbb{E}[y_i] = 0$. Taking expectation over the data yields

$$\mathbb{E}\big[\text{RMS}^2(\boldsymbol{y})\big] = \mathbb{E}\left[\frac{1}{d_{\text{out}}} \sum_{i=1}^{d_{\text{out}}} y_i^2\right] = \mathbb{E}[y_i^2] = \text{Var}(y_i). \qquad (6)$$

Therefore, when the input RMS $s_{\text{in}}$ is kept unchanged, preserving the output RMS scale is equivalent to preserving the

---

[1]We thus treat the *gate* projection in the same way as the *up* projection under RMS-preserving expansion, and regard the resulting gate activation output as a $\Theta(1)$ multiplicative factor in expectation.

per-coordinate variance

$$\text{Var}(y_i) = d_{\text{in}}\,\sigma_w^2\sigma_x^2, \tag{7}$$

where $\sigma_w^2$ and $\sigma_x^2$ denote the shared variances of $w_{ij}$ and $x_j$, respectively. We derive Eq. (7) in Appendix A.1.

### 3.2.2. FAN-OUT EXPANSION

In fan-out expansion, the output dimension grows from $d_{\text{out}}$ to $d'_{\text{out}}\,(> d_{\text{out}})$ while keeping $d_{\text{in}}$ unchanged, denoted by

$$\boldsymbol{y}' = \boldsymbol{W}'\boldsymbol{x}, \quad \boldsymbol{W}' = \begin{bmatrix} \boldsymbol{W} \\ \tilde{\boldsymbol{W}} \end{bmatrix} \in \mathbb{R}^{d'_{\text{out}} \times d_{\text{in}}}, \tag{8}$$

$$\boldsymbol{y}' = \begin{bmatrix} \boldsymbol{y} \\ \tilde{\boldsymbol{y}} \end{bmatrix} \in \mathbb{R}^{d'_{\text{out}}}, \quad \tilde{\boldsymbol{y}} = \tilde{\boldsymbol{W}}\boldsymbol{x}. \tag{9}$$

Naturally, fan-out expansion preserves activation RMS as long as the newly introduced output channels $\tilde{\boldsymbol{W}}$ (i.e., the new rows of $\boldsymbol{W}'$) are distributionally consistent with the pre-expansion ones. Concretely, when the added rows are initialized by *copying* or by *randomly sampling* from the same distribution as the original weights, the expanded output $\boldsymbol{y}'$ remains distributionally aligned with $\boldsymbol{y}$ and thus retains the same RMS scale.

### 3.2.3. FAN-IN EXPANSION

In fan-in expansion, the input dimension grows from $d_{\text{in}}$ to $d'_{\text{in}}\,(> d_{\text{in}})$ while keeping $d_{\text{out}}$ unchanged, denoted by

$$\boldsymbol{y}' = \boldsymbol{W}'\boldsymbol{x}', \quad \boldsymbol{W}' = \alpha \begin{bmatrix} \boldsymbol{W} & \tilde{\boldsymbol{W}} \end{bmatrix} \in \mathbb{R}^{d_{\text{out}} \times d'_{\text{in}}}, \tag{10}$$

$$\boldsymbol{x}' = \begin{bmatrix} \boldsymbol{x} \\ \tilde{\boldsymbol{x}} \end{bmatrix} \in \mathbb{R}^{d'_{\text{in}}}, \quad \boldsymbol{y}' = \alpha(\boldsymbol{W}\boldsymbol{x} + \tilde{\boldsymbol{W}}\tilde{\boldsymbol{x}}). \tag{11}$$

RMS-preserving fan-in expansion seeks the scaling factor $\alpha$ satisfying the invariance of Eq. (7) across expansion.

**Random or One-Side Copied.** If the newly added fan-in coordinates are initialized by same-distribution random sampling, or by copying on only one side (i.e., only $\boldsymbol{W}$ or only $\boldsymbol{x}$ is copied while the other remains random), then Eq. (6) and its underlying assumptions continue to hold, resulting in a shared per-coordinate variance after expansion:

$$\text{Var}(y_i') = \sum_{j=1}^{d'_{\text{in}}} \text{Var}(w'_{ij}x'_j) = \sum_{j=1}^{d'_{\text{in}}} \sigma_{w'}^2\sigma_{x'}^2 = d'_{\text{in}}\,\sigma_{w'}^2\sigma_{x'}^2. \tag{12}$$

With unchanged input scale $\sigma_{x'}^2 = \sigma_x^2$, variance preservation requires

$$d'_{\text{in}}\,\sigma_{w'}^2 = d_{\text{in}}\,\sigma_w^2 \implies \sigma_{w'} = \sqrt{\frac{d_{\text{in}}}{d'_{\text{in}}}}\,\sigma_w, \tag{13}$$

which implies that the weights should be rescaled as

$$w'_{ij} = \sqrt{\frac{d_{\text{in}}}{d'_{\text{in}}}}\,w_{ij}, \quad \forall i = 1,\ldots,d_{\text{out}},\ j = 1,\ldots,d'_{\text{in}}, \tag{14}$$

thereby keeping $\text{Var}(y_i') = \text{Var}(y_i)$ and the output RMS invariant under fan-in expansion.

**Both-Sides Copied.** A qualitatively different regime arises when *both* sides of the newly introduced fan-in coordinates are created by copying existing dimensions, where the independence across fan-in dimensions is violated.

Let $c$ denote the copy ratio. $0 < c \le 1$ corresponds to the setting where each copied dimension is duplicated exactly once, while $c > 1$ corresponds to that some dimensions may be copied multiple times. Generally, we have

$$d'_{\text{in}} = (1 + c)d_{\text{in}}. \tag{15}$$

The invariance of Eq. (7) requires the weights to be rescaled as

$$w'_{ij} = \begin{cases} \frac{1}{\sqrt{1+3c}}w_{ij}, & 0 < c \le 1, \\ \frac{1}{1+c}w_{ij}, & c > 1, \end{cases} \tag{16}$$

or equivalently,

$$w'_{ij} = \begin{cases} \sqrt{\frac{d_{\text{in}}}{3d'_{\text{in}}-2d_{\text{in}}}}\,w_{ij}, & d_{\text{in}} < d'_{\text{in}} \le 2d_{\text{in}}, \\ \frac{d_{\text{in}}}{d'_{\text{in}}}\,w_{ij}, & d'_{\text{in}} > 2d_{\text{in}}, \end{cases} \tag{17}$$

$\forall i = 1,\ldots,d_{\text{out}},\ j = 1,\ldots,d'_{\text{in}}$. We provide the full derivation of the above scaling factor in Appendix A.2.

**One-Side Zero.** Empirically, we find that RMS-preserving expansion should treat the zero-initialized side as *random* rather than strictly loss preserving at the expansion moment, and we include detailed analysis in Appendix B.

### 3.2.4. RMSNORM WEIGHT EXPANSION

We next discuss how to expand the RMSNorm scale, which is invoked only when the hidden dimension is expanded. For RMSNorm parameterized by $\boldsymbol{\gamma} \in \mathbb{R}^d$, omitting the $\epsilon$ term for clarity, we have

$$\boldsymbol{z} = \text{RMSNorm}(\boldsymbol{x}; \boldsymbol{\gamma}) = \frac{\boldsymbol{x} \odot \boldsymbol{\gamma}}{\text{RMS}(\boldsymbol{x})}, \quad z_i = \frac{x_i\gamma_i}{\text{RMS}(\boldsymbol{x})}. \tag{18}$$

Applying Eq. (6) to $\boldsymbol{z}$ yields

$$\mathbb{E}[\text{RMS}^2(\boldsymbol{z})] = \text{Var}(z_i) = \frac{1}{\text{RMS}^2(\boldsymbol{x})}\,\sigma_x^2\,\sigma_\gamma^2\ \sim\ \sigma_\gamma^2, \tag{19}$$

where $\sigma_x^2 := \text{Var}(x_i)$ and $\sigma_\gamma^2 := \text{Var}(\gamma_i)$, and Eq. (6) along with its underlying assumptions is used for both $\boldsymbol{x}$ and $\boldsymbol{z}$.

Therefore, preserving the output RMS of $\boldsymbol{z}$ under width expansion is effectively equivalent to preserving the RMS of parameter $\boldsymbol{\gamma}$. Thus, when expanding RMSNorm from $d$ to $d' > d$, initializing the new coordinates of $\boldsymbol{\gamma}$ by copying or randomly sampling from the same distribution naturally maintains $\text{RMS}(\boldsymbol{z})$ without any additional rescaling.

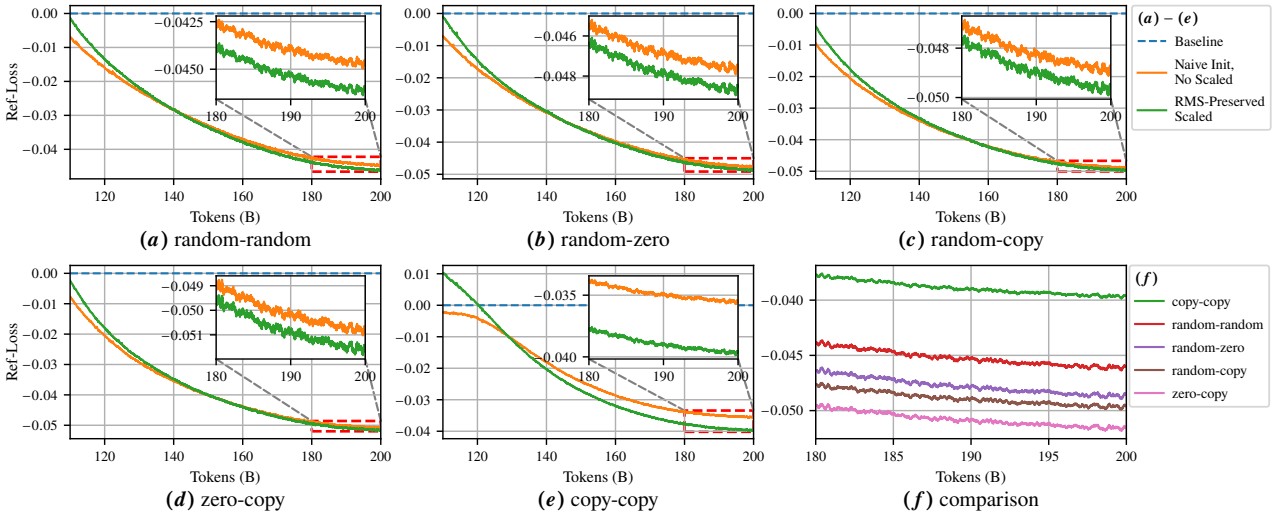

*Figure 1.* **RMS-preserving rescaling consistently improves late-stage convergence under MoE expert inner-dimension expansion.** We expand the expert inner dimension from $512 \to 1024$ at 100B tokens and plot *reference-loss* (relative to the pre-expansion reference) over the remaining training tokens. (*a*)–(*e*) sweep five (*up_proj – down_proj init*) pairs. In every case, *Naive Init, No Scaled* yields a smaller immediate loss gap, while *RMS-Preserved Scaled* overtakes later and converges to a lower final loss. (*f*) compares the RMS-preserved late-stage results and highlights a notable pattern: *both-sides copied* significantly underperforms other RMS-preserved strategies.

An ablation in Appendix C confirms that copy and random yield nearly identical final loss, since the RMSNorm parameter count is negligible relative to the linear layers. Therefore, in all our hidden-dimension expansion experiments such as those in Appendix E, we adopt copy initialization for the RMSNorm weights.

### 3.3. RMS-Preserving Expansion Improves Late-Stage Convergence.

**Experimental Setup.** We conduct progressive-learning experiments on OLMoE (Muennighoff et al., 2025) with 0.5 B active parameters and 2.5 B total parameters, trained for 200 B tokens in total using AdamW optimizer. We perform a mid-stage width expansion at 100 B tokens and then continue to train the expanded model for the remaining 100 B tokens under the same training recipe. We retain the original OLMoE pre-norm configuration throughout all our experiments to enable a controlled comparison against the established baseline. Details of the experimental setup are provided in Appendix D.

We consider two width-growth axes: (i) *Inner 2×*, which doubles the MoE expert intermediate dimension from 512 to 1024, and (ii) *Hidden 2×*, which doubles the model hidden size from 1024 to 2048. For *Hidden 2×*, we decouple hidden dimension from the usual constraint $hidden\_dim = qhead\_num \times head\_dim$ and expand only the hidden dimension while leaving the head dimension and the numbers of QKV heads unchanged. In each setting, we compare two initialization strategies for the newly introduced channels: (1) *Naive Init, No Scaled*, which applies copy/random/zero

initialization without any rescaling, and (2) *RMS-Preserved Scaled*, which applies the rescaling derived in Sec. 3.2 to enforce activation RMS consistency at the expansion moment.

**Results.** Fig. 1 reports expert-inner expansion, enumerating five fan-out/fan-in initialization pairs within each expert MLP, denoted as *up_proj – down_proj init*.

Across all initializations, *Naive Init, No Scaled* consistently yields a smaller immediate loss gap but worse late-stage convergence, whereas *RMS-Preserved Scaled* recovers steadily and converges to a lower final loss. The hidden-dimension expansion counterpart in Appendix E shows the same pattern. Overall, RMS-preserving expansion robustly improves late-stage convergence under both expert-inner and hidden-dimension growth across diverse initialization strategies.

Fig. 1(*f*) aggregates late-stage performance across initialization pairs under *RMS-Preserved Scaled*. While broadly beneficial, the *both-sides copied* configuration still significantly underperforms the other RMS-preserved variants. Notably, it also highlights that the magnitude of the immediate post-expansion loss spike is not predictive of final convergence. We provide a detailed analysis of the relationship between expansion-induced perturbation and final loss in Appendix F.

## 4. Breaking the Symmetry Lock

The experimental results in Sec. 3.3 reveal a counter-intuitive phenomenon: although the copy strategy strictly preserves the forward output at expansion, it consistently underperforms other RMS-preserving initializations, ex-

hibiting both slower post-expansion recovery and a higher eventual loss. Intuitively, copy-based initialization seems ideal, as it ensures a seamless loss transition and thus the most stable starting point. We argue that the gap is instead governed by a copy-induced backward-pass symmetry: duplicated components receive identical gradients and thus evolve identically, failing to diversify into distinct features and rendering the expanded capacity functionally redundant. We formally derive this mechanism with the following analysis in Sec. 4.1, and solve it by the asymmetric interventions developed in Sec. 4.2, converting it into the best initialization. This is why SPARKLING ultimately defaults to copy-copy initialization.

## 4.1. Identical Gradients Under Copy Expansion

Consider the linear layer in Eq. (2). We analyze gradient dynamics under $2\times$ width expansion with copy initialization.

**Fan-Out Expansion.** Specializing Eq. (8) to copy initialization gives $\tilde{W} = W$ and $W' = [W^\top, W^\top]^\top$, hence $y' = [y, y]$. If subsequent layers are also copied, back-propagation maintains symmetry: $\frac{\partial \mathcal{L}}{\partial y'} = [g, g]$ with $g = \frac{\partial \mathcal{L}}{\partial y}$. The gradient w.r.t. the expanded weights is:

$$\nabla_{W'}\mathcal{L} = \frac{\partial \mathcal{L}}{\partial y'}x^\top = \begin{bmatrix} g \\ g \end{bmatrix} x^\top = \begin{bmatrix} gx^\top \\ gx^\top \end{bmatrix}. \quad (20)$$

We provide the analogous analysis for fan-in expansion in Appendix A.3.

**Symmetry Lock.** In both cases, $\nabla_W\mathcal{L} = \nabla_{\tilde{W}}\mathcal{L}$ holds, indicating identical gradients in copy expansion. With symmetrically initialized optimizer states, i.e., identical momentum for AdamW, the two blocks receive identical updates, enforcing $W(t) = \tilde{W}(t)$ throughout training. This creates a "symmetry lock": despite increased parameters, the model remains in the original lower-dimensional subspace. The expanded neurons fail to learn distinct features, making width scaling inefficient unless the symmetry is explicitly broken.

**Orthogonalization Fails to Break Symmetry.** Advanced optimizers like Muon attempt to decorrelate updates by applying Newton-Schulz orthogonalization to the matrix-valued momentum, yet this mechanism fails to break the symmetry under copy-based expansion. Importantly, this step is typically implemented as a polynomial map of the Gram matrix: for a matrix $X_k$, the next iterate can be written in the generic form

$$X_{k+1} = X_k \phi(X_k^\top X_k), \quad (21)$$

where $\phi(\cdot)$ is a matrix polynomial corresponding to $\phi(G) = \frac{1}{2}(3I - G)$ or higher-order variants $\phi(G) = \alpha I + \beta G + \gamma G^2$ with appropriate coefficients (Jordan et al., 2024).

Consider the column-duplicated (fan-in) case where the momentum is initialized as $X_0 = [A_0, A_0]$. Let $P_k = $ $A_k^\top A_k$. Then the Gram matrix remains block-constant:

$$X_k^\top X_k = \begin{bmatrix} A_k^\top \\ A_k^\top \end{bmatrix} \begin{bmatrix} A_k, A_k \end{bmatrix} = \begin{bmatrix} P_k & P_k \\ P_k & P_k \end{bmatrix}. \quad (22)$$

Thus, applying the generic orthogonalization update yields

$$X_{k+1} = \begin{bmatrix} A_k, A_k \end{bmatrix} \phi\left(\begin{bmatrix} P_k & P_k \\ P_k & P_k \end{bmatrix}\right) := \begin{bmatrix} A_{k+1}, A_{k+1} \end{bmatrix}. \quad (23)$$

Since $\phi(\cdot)$ preserves block-exchange symmetry, the update in Eq. (23) retains two identical column blocks. Therefore, the orthogonalization step cannot spontaneously break the symmetry lock induced by copy initialization.

## 4.2. Breaking Symmetry in Practice

### 4.2.1. OPTIMIZER STATE RESET AS A NECESSARY INTERVENTION

Copy-based expansion yields identical gradients for the original and duplicated parameters. If the optimizer states are also initialized symmetrically—either by copying the existing states or by resetting all states to zero—the two halves receive identical updates, so the symmetry lock persists under both AdamW and Muon. To break this coupling symmetry without discarding the original model's training signal, we enforce an *asymmetric* treatment: retaining the optimizer states for the original $W$ and resetting the states for the new parameters $\tilde{W}$. Formally, the corresponding optimizer state matrix $S'$ (representing both first and second momentum for AdamW and momentum for Muon) is initialized as

$$S' = [S, 0], \quad (24)$$

where $S$ is the pre-expansion state of $W$ and $0$ initializes the state of $\tilde{W}$.

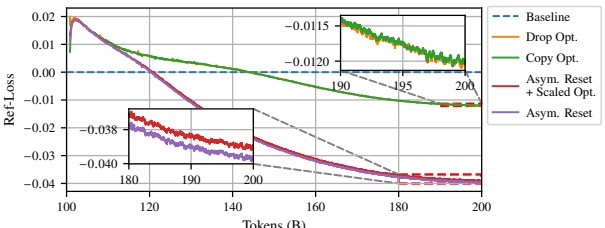

*Figure 2.* Optimizer-state handling under copy-based expansion. Symmetric treatments (*Drop/Copy*) exhibit a symmetry lock, yielding slower recovery and higher loss. Our asymmetric reset avoids this bottleneck, while state scaling provides no additional gain.

**Experimental setup.** Following Sec. 3.3, we study expert-inner expansion under copy-copy initialization and vary only the optimizer state handling at expansion, while applying RMS-preserving parameter scaling in all variants to keep forward activation scales consistent. We compare four treatments: (i) *Drop Opt.*, globally reset all states, (ii) *Copy*

*Opt.*, duplicate states, (iii) *Asymmetric Reset*, reset states only for new channels, and (iv) *Asymmetric Reset + Scaled Opt.*, which additionally applies the exact same parameter scaling to the optimizer states, ensuring that parameters and their associated optimizer states are rescaled consistently, following the optimizer state scaling of Shen et al. (2022). The first two are symmetric, whereas the latter two break symmetry via optimizer dynamics.

**Results.** Fig. 2 reveals a clear gap between symmetric and asymmetric treatments. Both symmetric baselines (*Drop Opt.* and *Copy Opt.*) underperform substantially, showing slower post-expansion recovery and a higher converged loss, consistent with copy-induced backward symmetry that keeps duplicated components tightly coupled. In contrast, *Asymmetric Reset* improves both recovery speed and final loss, indicating that resetting optimizer states only for the new channels suffices to break the symmetry lock and enable feature diversification. Notably, explicit optimizer state scaling (*Asymmetric Reset + Scaled Opt.*) yields no additional gain, suggesting that the strict alignment of state-parameter scaling does not appear to be a critical requirement for the width expansion regimes considered here. Any initial misscaling is quickly corrected by subsequent gradient updates.

### 4.2.2. ASYMMETRIC LEARNING RATE RE-WARMUP

For training from scratch, we use a standard cosine decay learning rate scheduler with linear warmup. Let $T_w$ denote the number of re-warmup steps, $T$ the total number of steps, and let $\eta_0, \eta_{\max}$ and $\eta_{\min}$ be the initial, peak and final learning rates, respectively. The baseline schedule is

$$\eta(t) = f\left(t; T_w, T, \eta_0, \eta_{\max}, \eta_{\min}\right)$$

$$= \begin{cases} \eta_0 + (\eta_{\max} - \eta_0) \cdot \dfrac{t}{T_w}, & 0 \le t < T_w, \\[2mm] \eta_{\min} + (\eta_{\max} - \eta_{\min})\,\psi\left(\dfrac{t - T_w}{T - T_w}\right), & T_w \le t \le T, \end{cases}$$

$$\tag{25}$$

where

$$\psi(x) = \frac{1}{2}\left(1 + \cos(\pi x)\right). \tag{26}$$

At an expansion point $t_e$, we keep the original parameters on the same baseline schedule to preserve continuity, i.e., $\eta(t) = f\left(t; T_w, T, \eta_0, \eta_{\max}, \eta_{\min}\right)$ for all $t$.

For the newly introduced parameters, we perform an asymmetric re-warmup that starts exactly from the current learning rate $\eta_e = \eta(t_e)$ and warms up for $\tau_w$ steps to a new peak learning rate proportional to $\eta_e$:

$$\hat{\eta}_{\max} = \rho \cdot \eta_e, \quad \eta_e = \eta(t_e), \tag{27}$$

where $\rho$ is the re-warmup ratio. The learning rate for the new parameters is then defined as

$$\eta_{\text{new}}(t) = f(t - t_e; \tau_w, T - t_e, \eta_e, \hat{\eta}_{\max}, \eta_{\min}), \quad t > t_e, \tag{28}$$

where, after rewarmup, the schedule follows the same cosine-decay regime and decays to the shared minimum learning rate $\eta_{\min}$. See Appendix G for a sample curve.

### 4.3. Asymmetric Learning Rate Re-warmup Further Improves Convergence Consistently.

**Experimental setup.** Following Sec. 3.3, we evaluate *asymmetric learning rate re-warmup* across different width-expansion axes. We consider three expansion settings: expert-inner (*Inner 2×*), hidden-dimension (*Hidden 2×*), and joint (*Hidden 2× & Inner 2×*). For all settings, we apply RMS-preserving scaling and asymmetric optimizer state reset, then ablate re-warmup by comparing runs with vs. without it. We report both copy-copy and zero-copy initializations (the best-performing no-re-warmup setting in our earlier analysis, see Fig. 1) to assess robustness. We set the re-warmup ratio $\rho = 1.3$ and the number of re-warmup steps $\tau_w = 250$ based on Appendix H.

**Results.** Fig. 3 shows that asymmetric learning rate re-warmup consistently improves convergence across width axes and initialization strategies. Across all width axes in three settings, enabling re-warmup yields a lower eventual loss under the same token budget. The benefit holds for both zero-copy, where new channels begin with near-zero forward contribution, and copy-copy, which strictly preserves the forward mapping.

Notably, the gain is largest for copy-copy: re-warmup closes the post-expansion gap to zero-copy, and reaches the lowest final loss among variants. This is consistent with our symmetry-lock analysis in Sec. 4.1: beyond RMS preservation and asymmetric state reset, re-warmup injects controlled optimization asymmetry that encourages the duplicated subspaces to diversify into effective capacity rather than remaining redundant copies. It is therefore a robust component of SPARKLING, reliably improving convergence across width-expansion axes and initialization regimes.

## 5. Discussions

Taken together, our SPARKLING framework comprises (i) RMS-preserving scaling, (ii) copy-based initialization, (iii) asymmetric optimizer state reset, and (iv) an asymmetric learning rate re-warmup schedule. In this section, we evaluate its overall performance.

### 5.1. Overall Downstream Performance

**Experimental setup.** Following Sec. 4.3, we further evaluate downstream performance under three 2× width-expansion settings. For each setting, we compare (i) the *Baseline (small)* model before expansion, (ii) the *Baseline (expand)* model trained from scratch at the target width un-

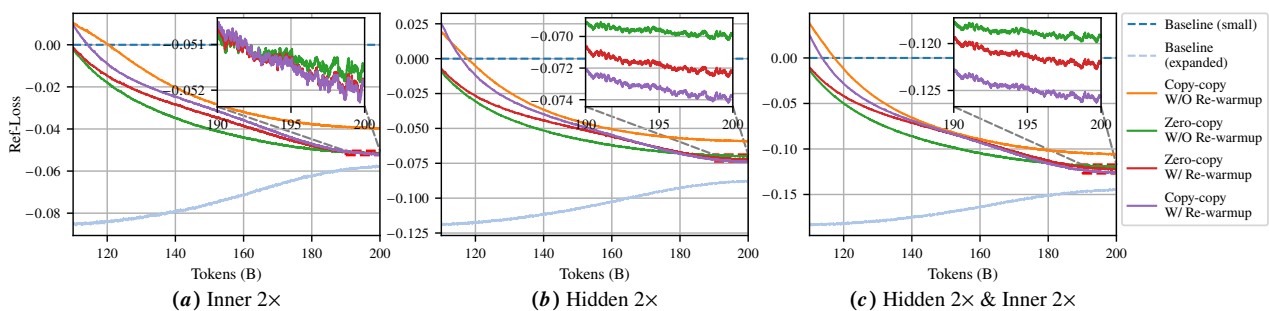

*Figure 3.* **Asymmetric re-warmup consistently improves convergence under mid-stage width expansion.** Across Inner $2\times$, Hidden $2\times$, and joint expansion, re-warmup lowers the final loss for both RMS-preserving copy-copy and zero-copy. Copy-copy benefits most, achieving the best final loss, effectively mitigating copy-induced symmetry lock.

*Table 1.* **Downstream performance under $2\times$ mid-stage width expansion.** Across Inner $2\times$, Hidden $2\times$, and joint expansion, SPARKLING matches or outperforms the from-scratch expanded baseline on most tasks and achieves the best average, despite a slightly higher final pre-training loss.

| Model | Loss (↓) | ARC-C (↑) | ARC-E (↑) | Arith. (↑) | BoolQ (↑) | CSQA (↑) | HellaS. (↑) | MMLU (↑) | OBQA (↑) | PIQA (↑) | SciQ (↑) | SIQA (↑) | WinoG. (↑) | Avg. (↑) |
|---|---|---|---|---|---|---|---|---|---|---|---|---|---|---|
| Baseline (small) | 2.3673 | 41.47 | 72.46 | 43.63 | 66.36 | 47.58 | 65.86 | 32.75 | 39.40 | 76.28 | 92.50 | 46.57 | 62.19 | 57.26 |
| *Inner 2×* | | | | | | | | | | | | | | |
| Baseline (expand) | **2.3096** | 43.14 | **74.56** | 55.67 | **67.80** | 47.58 | **69.45** | 32.67 | **42.40** | 78.18 | 92.80 | 46.67 | 64.80 | 59.64 |
| Naive FP scaled | 2.3276 | **44.15** | 73.86 | 51.10 | 66.45 | 48.24 | 68.04 | 32.71 | 41.80 | 77.09 | 92.90 | 46.98 | 64.25 | 58.96 |
| SPARKLING | 2.3153 | 43.48 | 74.39 | **60.50** | 66.82 | **49.06** | 69.21 | **34.05** | 41.60 | **78.35** | **93.30** | **47.80** | **65.04** | **60.30** |
| *Hidden 2×* | | | | | | | | | | | | | | |
| Baseline (expand) | **2.2795** | **44.48** | **77.72** | 54.47 | 67.06 | 48.32 | **70.66** | 33.82 | 42.00 | **78.56** | **93.90** | 47.85 | **66.61** | 60.46 |
| Naive FP scaled | 2.3082 | 41.81 | 73.86 | 53.77 | **67.37** | 49.06 | 69.40 | 32.00 | 41.00 | 78.18 | 93.00 | 47.44 | 64.17 | 59.25 |
| SPARKLING | 2.2933 | **44.48** | 76.14 | **61.90** | 67.16 | **49.80** | 70.06 | **34.49** | **43.40** | 78.45 | 93.40 | **48.26** | 65.98 | **61.13** |
| *Hidden 2× & Inner 2×* | | | | | | | | | | | | | | |
| Baseline (expand) | **2.2225** | **46.82** | 76.14 | 55.03 | 67.65 | 49.71 | 72.54 | 33.20 | **43.80** | 79.00 | 93.30 | **48.57** | 64.88 | 60.89 |
| Naive FP scaled | 2.2615 | 45.82 | 74.56 | 58.67 | 67.09 | **51.60** | 71.92 | 33.64 | 41.40 | 79.00 | 93.70 | 47.54 | 65.75 | 60.89 |
| SPARKLING | 2.2415 | **46.82** | **77.19** | **66.00** | **69.08** | 50.86 | **72.82** | **35.07** | 43.00 | **79.11** | **94.10** | 48.36 | **68.19** | **62.55** |

*Table 2.* Compute-cost comparison under a fixed token budget.

| Method | Total Tokens | Expand @ | Act./Tot. Params | FLOPs (×10²⁰) | Wall-clock (h) | FLOPs Saved | Speed-up |
|---|---|---|---|---|---|---|---|
| Baseline | 200B | - | 450M/2.56B | 5.40 | 48 | | |
| *Inner 2×* | | | | | | | |
| From Scratch | 200B | - | 751M/5B | 9.01 | 84 | - | - |
| SPARKLING | | 100B | | **7.21** | **66** | 20% | 1.27× |
| *Hidden 2×* | | | | | | | |
| From Scratch | 200B | - | 900M/5.13B | 10.80 | 96 | - | - |
| SPARKLING | | 100B | | **8.10** | **75** | 25% | 1.29× |
| *Hidden 2× & Inner 2×* | | | | | | | |
| From Scratch | 200B | - | 1.5B/9.96B | 18.00 | 209 | - | - |
| SPARKLING | | 100B | | **11.70** | **140** | 35% | 1.49× |

der the same token budget, (iii) a *Naive function-preserving scaled* variant with copy-based initialization but without our interventions, and (iv) our *SPARKLING*, which combines RMS-preserving scaling with asymmetric optimizer state reset and asymmetric learning rate re-warmup for newly introduced parameters. We report the final pre-training loss and downstream accuracies, including ARC-C/E (Clark et al., 2018), Arithmetic (Brown et al., 2020), BoolQ (Clark et al., 2019), CommonsenseQA (Talmor et al., 2019), HellaSwag (Zellers et al., 2019), MMLU (Hendrycks et al., 2021), OpenBookQA (Mihaylov et al., 2018), PIQA (Bisk et al., 2020), SciQ (Welbl et al., 2017), SocialIQA (Sap et al., 2019), Winogrande (Sakaguchi et al., 2020).

**Results.** Table 1 shows that mid-stage expansion still leaves a small gap in final pre-training loss relative to training the expanded model from scratch. Nevertheless, SPARKLING achieves the best downstream average among the expansion variants and matches or exceeds the from-scratch expanded baseline on most tasks.

The residual loss gap is itself the expected behavior of training under reduced compute. Following the power-law $L \propto C^{-\alpha}$ of Kaplan et al. (2020), our compute saving naturally leads to a slightly higher absolute loss. We attribute the loss-downstream mismatch to expanded models entering different regions of the loss landscape than from-scratch counterparts, where the geometry favors better generalization.

Overall, these results validate the reliability of SPARKLING: despite a slightly larger final pre-training loss than the from-scratch counterpart, our framework consistently improves downstream performance across diverse width-expansion axes.

### 5.2. Ablations and Baseline Comparisons

**Ablation studies.** Appendix I isolates the contributions of the two principles underlying SPARKLING, i.e., *signal preservation* via RMS-preserved scaling and *symmetry*

*breaking* via the asymmetric strategies, and shows that they deliver complementary, *additive* gains on both final pre-training loss and downstream performance, with neither component alone matching the full framework.

**Comparison with prior works.** On final pre-training loss, SPARKLING outperforms both representative initialization-based heuristics (Chen et al., 2016; 2022; Du et al., 2024; Wang et al., 2024; Wu et al., 2020; Yuan et al., 2023; Wu et al., 2019) in Appendix J and dynamics-based strategies (Shen et al., 2022; Wang et al., 2024; Yuan et al., 2023) in Appendix K. This advantage further carries over to downstream performance: against these baselines, SPARKLING attains the strongest downstream average in Appendix L, confirming that it delivers consistent gains both on pre-training loss and downstream performance.

### 5.3. Generality across Optimizers, Architectures, and Stages

Beyond the MoE-with-AdamW setting studied above, we further validate that SPARKLING generalizes along three orthogonal axes. (i) *Optimizer families*: it remains effective under the spectral-style Muon optimizer, where both RMS-preserved scaling and asymmetric re-warmup continue to lower the final loss (Appendix M), showing SPARKLING's generality across optimizer families. (ii) *Architectures*: the same recipe transfers to dense models, matching or exceeding the from-scratch baseline on most downstream tasks (Appendix N). (iii) *Expansion stages*: SPARKLING generalizes naturally to iterative multi-stage expansion (e.g., $256 \rightarrow 512 \rightarrow 1024$), retaining its effectiveness at each stage while further reducing total compute (Appendix O).

### 5.4. Iso-token Compute Savings

Now that the effectiveness of our expansion framework has been validated, we finally return to the core motivation of progressive learning—reducing training costs while retaining or even surpassing the performance of the target-width model. We quantify the computational savings by comparing mid-stage width expansion with training the target-width model from scratch *under the same training token budget*. Following Kaplan et al. (2020), we approximate pre-training compute as $C \approx 6ND$, where $N$ is the number of *active* parameters and $D$ is the total training tokens. Suppose that the expansion occurs at $D_e$ tokens, where the small model with active size $N_{\text{small}}$ is trained for $D_e$ tokens, and the expanded model of active size $N_{\text{large}}$ is trained for $(D - D_e)$ tokens, yielding

$$C^* \approx 6\big(N_{\text{small}}D_e + N_{\text{large}}(D - D_e)\big), \qquad (29)$$

whereas training the expanded model from scratch costs $C_{\text{scratch}} \approx 6N_{\text{large}}D$. We report the relative reduction as *FLOPs Saved* $= 1 - C^*/C_{\text{scratch}}$, and the empirical wall-clock *Speed-up* as $T_{\text{scratch}}/T^*$.

Table 2 summarizes results across three $2\times$ width-expansion settings. Under the same 200 B-token budget, SPARKLING saves $20\,\%$–$35\,\%$ training FLOPs relative to training the expanded model from scratch, and achieves up to a $1.49\times$ measured wall-clock speed-up under $2\times$ width expansion. Overall, SPARKLING matches or even exceeds the performance of the from-scratch expanded model while substantially reducing training costs, making mid-stage width expansion practically advantageous.

### 5.5. Iso-compute Scalability

To probe scalability beyond the iso-token setting, we further provide an iso-compute analysis in Appendix P. Given the same total compute budget for both SPARKLING and the from-scratch baseline, SPARKLING consistently achieves a lower final loss than the from-scratch baseline across all three MoE width axes, and the same advantage holds on a dense architecture, indicating a more favorable compute–loss scaling behavior with a larger scaling exponent $\alpha$ in $L \propto C^{-\alpha}$.

This iso-compute advantage further strengthens under iterative expansion: a 2-stage expert-inner expansion $256 \rightarrow 512 \rightarrow 1024$ yields a larger iso-compute loss reduction than the 1-stage variant while further reducing total compute, showing that SPARKLING continues to translate width into effective capacity at matched compute under multi-stage expansion.

## 6. Conclusion and Future Work

We proposed SPARKLING, a systematic progressive learning framework via width expansion, and resolved the challenges arising during mid-stage model expansion. In contrast to conventional function-preserving perspectives, we emphasize signal preservation by maintaining the RMS scale of activations during expansion. To break the symmetry induced by copy-based initialization, we apply asymmetric optimizer state reset together with asymmetric learning rate re-warmup. Across dense and MoE architectures, multiple width axes, optimizer families, and multiple expansion stages, extensive experiments validate both the effectiveness and the efficiency of our framework.

While our results are promising, several avenues remain for future exploration. First, a unified principle of simultaneous width and depth expansion has yet to be established. Moreover, we aim to investigate whether our RMS preservation strategy could satisfy the $\mu$P condition (Yang et al., 2021), where the transferability of optimal hyperparameters is naturally ensured after expansion. We view these as critical future work toward developing a more comprehensive, "tuning-free" framework for progressive learning.

## Acknowledgements

DH is supported by National Science Foundation of China (NSFC62376007), National Science Foundation of China (under Key Project No. 92570203), Beijing Natural Science Foundation (Z250001) and Beijing Major Science and Technology Project under Contract no. Z251100008425004.

## Impact Statement

This paper presents work whose goal is to advance the field of Machine Learning. There are many potential societal consequences of our work, none which we feel must be specifically highlighted here.

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

# A. Derivations

## A.1. Eq. (7): The Per-Coordinate Variance Under Fan-In Aggregation

We provide the intermediate steps used in Sec. 3.2.1 to derive RMS preservation for a fan-in variance constraint.

Consider the linear layer $\boldsymbol{y} = \boldsymbol{W}\boldsymbol{x}$ with $\boldsymbol{W} \in \mathbb{R}^{d_{\text{out}} \times d_{\text{in}}}$ and $\boldsymbol{x} \in \mathbb{R}^{d_{\text{in}}}$. For a fixed output coordinate $i \in \{1, \ldots, d_{\text{out}}\}$, we write

$$y_i = \sum_{j=1}^{d_{\text{in}}} w_{ij} x_j. \tag{30}$$

Assume that, across the fan-in dimensions, the pairs $\{(w_{ij}, x_j)\}_{j=1}^{d_{\text{in}}}$ are mutually independent, and that $\boldsymbol{W}$ and $\boldsymbol{x}$ are independent of each other. Under these conditions, the variance of $y_i$ decomposes additively:

$$\text{Var}(y_i) = \text{Var}\left(\sum_{j=1}^{d_{\text{in}}} w_{ij} x_j\right) = \sum_{j=1}^{d_{\text{in}}} \text{Var}(w_{ij} x_j). \tag{31}$$

If, in addition, the fan-in terms are homoscedastic and centered in the sense that $\mathbb{E}[w_{ij}] = \mathbb{E}[x_j] = 0$ and $\text{Var}(w_{ij}) = \sigma_w^2$, $\text{Var}(x_j) = \sigma_x^2$ for all $j$, then each product term shares the same variance $\text{Var}(w_{ij} x_j) = \sigma_w^2 \sigma_x^2$. Substituting this into Eq. (31) yields

$$\text{Var}(y_i) = \sum_{j=1}^{d_{\text{in}}} \sigma_w^2 \sigma_x^2 = d_{\text{in}}\, \sigma_w^2 \sigma_x^2, \tag{32}$$

which is the expression in Eq. (7) of the main text. Combined with Eq. (6), this shows that (when $s_{\text{in}}$ is fixed) preserving the output RMS scale is equivalent to preserving $\text{Var}(y_i)$, and under the above assumptions this reduces to keeping $d_{\text{in}}\sigma_w^2 \sigma_x^2$ invariant across the expansion.

## A.2. Eq. (16)–(17): RMS-Preserving Rescaling under Fan-In Expansion with Both-Sides Copied

A qualitatively different regime arises when *both* sides of the newly introduced fan-in coordinates are created by copying existing dimensions, i.e., the new columns of $\boldsymbol{W}'$ and the new coordinates of $\boldsymbol{x}'$ are both duplicated from the same subset of original fan-in dimensions, respectively. In this case, the independence across fan-in dimensions is violated: the copied pairs $(w'_{ij}, x'_j)$ are no longer independent replicas but perfectly correlated duplicates of some original terms. As a result, the variance no longer decomposes as a simple sum of $d'_{\text{in}}$ independent contributions, and the duplicated terms contribute quadratically through covariance.

Given Eq. (15) with $c$ denoting the copy ratio, we first consider the setting $0 < c \leq 1$ where each copied dimension is duplicated exactly once. Let $\mathcal{R}$ be the set of copied indices with $|\mathcal{R}| = c\, d_{\text{in}}$, and $\mathcal{S}$ be the remaining indices with $|\mathcal{S}| = (1 - c)d_{\text{in}}$. Under one-to-one copying on both $\boldsymbol{W}$ and $\boldsymbol{x}$, each duplicated dimension contributes twice with identical value, yielding

$$y'_i = \sum_{j=1}^{d'_{\text{in}}} w'_{ij} x'_j = 2 \sum_{r \in \mathcal{R}} w'_{ir} x_r + \sum_{s \in \mathcal{S}} w'_{is} x_s. \tag{33}$$

Under the same independent assumptions as above on the *original* terms, the variance of $y'_i$ becomes

$$\begin{aligned}
\text{Var}(y'_i) &= \text{Var}\left(2 \sum_{r \in \mathcal{R}} w'_{ir} x_r + \sum_{s \in \mathcal{S}} w'_{is} x_s\right) \\
&= 4 \sum_{r \in \mathcal{R}} \text{Var}(w'_{ir} x_r) + \sum_{s \in \mathcal{S}} \text{Var}(w'_{is} x_s) \\
&= 4|\mathcal{R}|\sigma_{w'}^2 \sigma_x^2 + |\mathcal{S}|\sigma_{w'}^2 \sigma_x^2 \\
&= \left(4c\, d_{\text{in}} + (1 - c)d_{\text{in}}\right)\sigma_{w'}^2 \sigma_x^2 \\
&= d_{\text{in}}(1 + 3c)\sigma_{w'}^2 \sigma_x^2.
\end{aligned} \tag{34}$$

When the input scale is kept unchanged, preserving the original variance requires rescaling the weights in the expanded layer. Let $\sigma_{w'}^2$ denote the post-rescaling weight variance; enforcing $\mathrm{Var}(y_i') = \mathrm{Var}(y_i)$ gives

$$d_{\mathrm{in}}(1+3c)\,\sigma_{w'}^2\sigma_x^2 = d_{\mathrm{in}}\,\sigma_w^2\sigma_x^2 \implies \sigma_{w'}^2 = \frac{1}{1+3c}\sigma_w^2, \tag{35}$$

or equivalently,

$$w_{ij}' = \frac{1}{\sqrt{1+3c}}\,w_{ij}, \quad 0 < c \le 1, \quad \forall i = 1,\dots,d_{\mathrm{out}},\, j = 1,\dots,d_{\mathrm{in}}'. \tag{36}$$

For $c > 1$ where some dimensions might be copied multiple times, the variance of $y_i'$ becomes

$$\mathrm{Var}(y_i') = (1+c)^2 d_{\mathrm{in}}\sigma_{w'}^2\sigma_x^2. \tag{37}$$

When the input scale is kept unchanged, enforcing $\mathrm{Var}(y_i') = \mathrm{Var}(y_i)$ gives

$$(1+c)^2 d_{\mathrm{in}}\,\sigma_{w'}^2\sigma_x^2 = d_{\mathrm{in}}\,\sigma_w^2\sigma_x^2 \implies \sigma_{w'}^2 = \frac{1}{(1+c)^2}\sigma_w^2, \tag{38}$$

or equivalently,

$$w_{ij}' = \frac{1}{1+c}\,w_{ij}, \quad c > 1, \quad \forall i = 1,\dots,d_{\mathrm{out}},\, j = 1,\dots,d_{\mathrm{in}}'. \tag{39}$$

Combining Eqs. (36) and (39) yields the final rescaling rule in Eq. (16). Substituting $c = d_{\mathrm{in}}'/d_{\mathrm{in}} - 1$ into these equations gives the equivalent form in Eq. (17).

### A.3. Identical Gradients Under Copy Expansion for Fan-In Expansion

For the input expansion defined in Eq. (10), copy initialization sets $\tilde{W} = W$ such that $W' = \alpha[W, W]$, with the input duplicated as $x' = [x, x]$. The forward pass yields $y' = \alpha(Wx + Wx)$. During backward propagation:

$$\nabla_{W'}\mathcal{L} = \frac{\partial\mathcal{L}}{\partial y'}(x')^\top = g\left[x^\top, x^\top\right] = \left[gx^\top, gx^\top\right]. \tag{40}$$

This shows that the two copied blocks receive identical gradients and that the uniform scalar $\alpha$ does not affect this symmetry argument.

## B. RMS Scale Under Zero Initialization

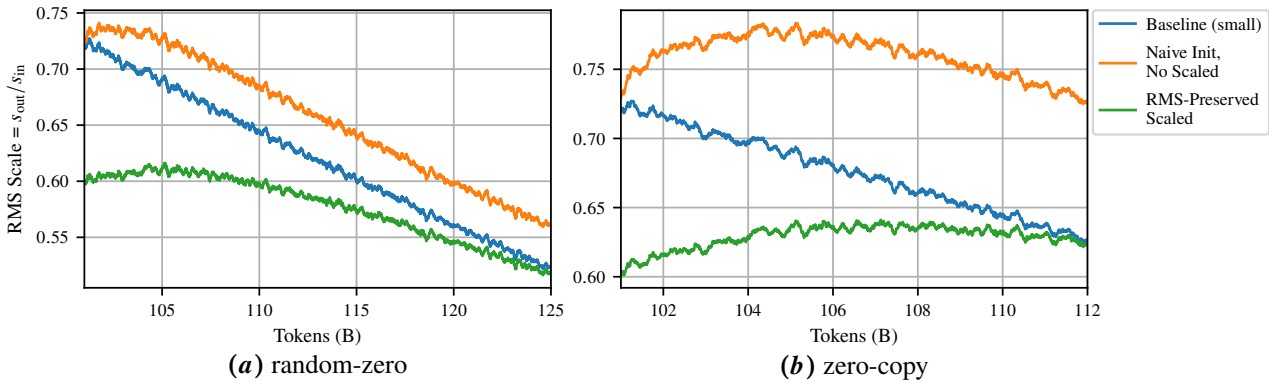

*Figure 4.* Under zero initialization, RMS-preserved scaling enables the post-expansion activation RMS scale to quickly recover toward the original baseline, indicating that zero initialization should be treated as random under RMS-preserving expansion.

In Fig. 4, we analyze a subtle but practically important corner case for RMS-preserving expansion: one-sided zero initialization. We consider two representative regimes, *random-zero* and *zero-copy*, and observe the RMS scale of the output and input activations of the whole MLP according to Eq. (3).

We empirically find that, when applying RMS-preserving scaling under zero initialization, the zero-initialized side should be treated as *random* rather than as a special perfectly *loss-preserving* case. Intuitively, a zero-initialized block becomes a gradient-driven random distribution after the very first update, so its effective statistics quickly resemble those of a randomly initialized block. While omitting RMS-preserving scaling can be strictly loss-preserving at the expansion moment and therefore naturally satisfies RMS-scale preservation at $t = t_e$, we observe that the RMS-preserving scaling variant that treats the zero side as random yields an activation RMS ratio that remains closer to the original baseline scale as post-expansion training proceeds. In contrast, the RMS scale under naive unscaled zero initialization drifts and does not exhibit a recovery trend toward the baseline. This behavior is consistent with the fact that zero initialization necessarily disrupts the pre-expansion parameter distribution and thus requires a nontrivial number of steps to re-enter a compatible statistical regime. Accordingly, our emphasis is on the RMS scale shape after the model has taken a small number of post-expansion updates, rather than on the degenerate preservation at the boundary itself.

## C. Ablation of RMSNorm Weight Expansion

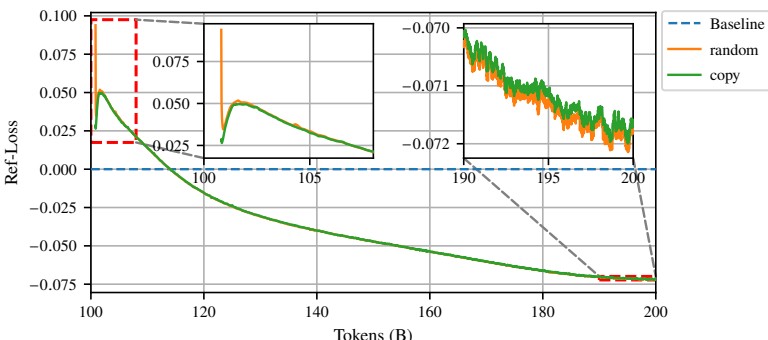

*Figure 5.* **Ablation of RMSNorm weight initialization.** Under hidden-dimension copy-copy expansion, random and copy initializations for RMSNorm weights quickly align in loss and maintain consistency until the end of training, achieving nearly identical final loss.

We ablate *copy* against *random* initialization for RMSNorm weights while keeping the rest of SPARKLING unchanged, including copy-copy initialization for linear layers, RMS-preserving scaling, and asymmetric strategies. As shown in Fig. 5, the two choices quickly align after expansion, remain consistent through training, and reach nearly identical final losses. We therefore use *copy* as the default RMSNorm initialization for all hidden-dimension expansion experiments and leave the main fan-in/fan-out ablation in Fig. 1 focused on linear-layer initializations.

## D. Detailed Experimental Setup

### D.1. Baseline Model Configuration

We list the detailed hyperparameters and architectural configuration of pre-expansion baseline model before expansion in Table 3. In addition to these settings, we adopt a pre-norm design by inserting RMSNorm before both the attention and MLP sublayers. Moreover, within the attention block, we apply per-head $q/k$ normalization along the head-dimension for each query and key head, i.e., normalizing the projected $q$ and $k$ vectors within each head over the $d_{\text{head}}$ dimension.

Notably, since we tie the word embedding and the output projection, hidden-size expansion makes the shared matrix act as fan-out on the embedding side but fan-in on the output side. As our RMS-preserving scaling requires different factors for these two roles, we compensate the fan-in factor by multiplying the corresponding coefficient after the final output projection for this special case.

### D.2. Training Hyperparameters

We summarize the training hyperparameters in Table 4. Unless otherwise specified (e.g., the Muon experiments in Sec. M), we optimize with AdamW using $(\beta_1, \beta_2) = (0.9, 0.95)$, $\epsilon = 10^{-8}$, and weight decay 0.1, where we apply weight decay to all parameters including norms and embeddings. We use a cosine learning rate schedule with a linear warmup over 3% of total steps, and decay to a minimum learning rate set by a final ratio of 0.01 relative to the peak learning rate. All experiments are run on a cluster of $64 \times$ NVIDIA A100 GPUs with $80\,\text{GB}$ memory each, using a global batch size of 768

with per-device microbatch size 3.

For all models trained from scratch, we set the peak learning rate to the step-law optimum reported by Li et al. (2025) and apply a batch-size scaling to obtain the corresponding value under our training setup in Table 5. In contrast, when expanding from a smaller model, we empirically find it more effective to keep the pre-expansion peak learning rate for training the expanded model with the same peak LR.

<table>
<tr><td colspan="2">*Table 3.* Baseline model Configuration.</td></tr>
<tr><td>**Configuration**</td><td>**Value**</td></tr>
<tr><td>Number of Hidden Layers ($L$)</td><td>24</td></tr>
<tr><td>Hidden Size ($d_{\text{model}}$)</td><td>1024</td></tr>
<tr><td>Expert Intermediate Size ($d_{\text{ffn}}$)</td><td>512</td></tr>
<tr><td>Number of Attention Heads ($n_{\text{heads}}$)</td><td>16</td></tr>
<tr><td>Number of Key/Value Heads ($n_{\text{kv}}$)</td><td>4</td></tr>
<tr><td>Head Dimension ($d_{\text{head}}$)</td><td>96</td></tr>
<tr><td>MoE Number of Experts ($E$)</td><td>64</td></tr>
<tr><td>MoE Top-$k$ ($k$)</td><td>8</td></tr>
<tr><td>Embedding Size ($|\mathcal{V}|$)</td><td>50304</td></tr>
<tr><td>Tie Word Embeddings</td><td>True</td></tr>
<tr><td>Activation Type</td><td>SwiGLU</td></tr>
<tr><td>Norm Type</td><td>RMSNorm</td></tr>
<tr><td></td><td>Pre-norm</td></tr>
<tr><td>Positional Embedding</td><td>RoPE</td></tr>
<tr><td>Use Bias</td><td>False</td></tr>
</table>

*Table 4.* Training hyperparameter configuration.

| Configuration | Value |
| --- | --- |
| Total Training Tokens | 200 B |
| Optimizer | AdamW |
| Peak Learning Rate | $1.9556 \times 10^{-3}$ |
| AdamW Betas ($\beta_1, \beta_2$) | $(0.9, \ 0.95)$ |
| AdamW Epsilon $\epsilon$ | $1.0 \times 10^{-8}$ |
| Weight Decay | 0.1 |
| Decay Norm & Bias | True |
| Decay Embeddings | True |
| LR Scheduler | Cosine w/ Warmup |
| Warmup Steps | 3% of total steps |
| Final LR Ratio | 0.01 |
| Max Sequence Length | 4096 |
| Global Batch Size | 768 |
| Device Microbatch Size | 3 |
| Number of GPUs | 64 |

*Table 5.* Step-law optimal learning rates (Li et al., 2025) across model sizes and the corresponding batch-size-scaled learning rates.

| Model | Total Tokens | Act. Params | Tot. Params | Step-law LR | Scaled LR |
| --- | --- | --- | --- | --- | --- |
| Baseline | 200B | 450M | 2.56B | 1.033e-3 | 1.449e-3 |
| Inner 2× | 200B | 751M | 5B | 6.410e-4 | 8.988e-4 |
| Hidden 2× | 200B | 900M | 5.13B | 5.760e-4 | 8.630e-4 |
| Hidden 2× & Inner 2× | 200B | 1.5B | 9.96B | 3.920e-4 | 5.497e-4 |

# E. Hidden-Dimension Expansion: RMS-Preserving Scaling

Following the experimental setting in Sec. 3.3, we provide the hidden-dimension counterpart of our RMS-scale analysis by doubling the model hidden size from 1024 to 2048 at 100 B tokens and continuing training to 200 B tokens under the same training recipe. Fig. 6 shows the same qualitative conclusion as expert-inner growth: while naive unscaled initialization can exhibit a smaller instantaneous loss discontinuity at the expansion moment, enforcing RMS-scale consistency via our RMS-preserving rescaling yields consistently better late-stage recovery and a lower converged loss across initialization regimes.

# F. Analysis between Perturbation and Final Loss

Under expert-inner expansion, Fig. 7 compares the immediate post-expansion loss spike with the final pre-training loss across fan-out/fan-in initialization pairs. The spike size does not directly predict final convergence. For example, random-copy has a larger initial perturbation than random-zero but reaches a lower final loss. This suggests that strict function preservation is neither necessary nor sufficient for expansion. Instead, the post-expansion training dynamics matter more, motivating our focus on RMS-scale consistency in Sec. 3.3.

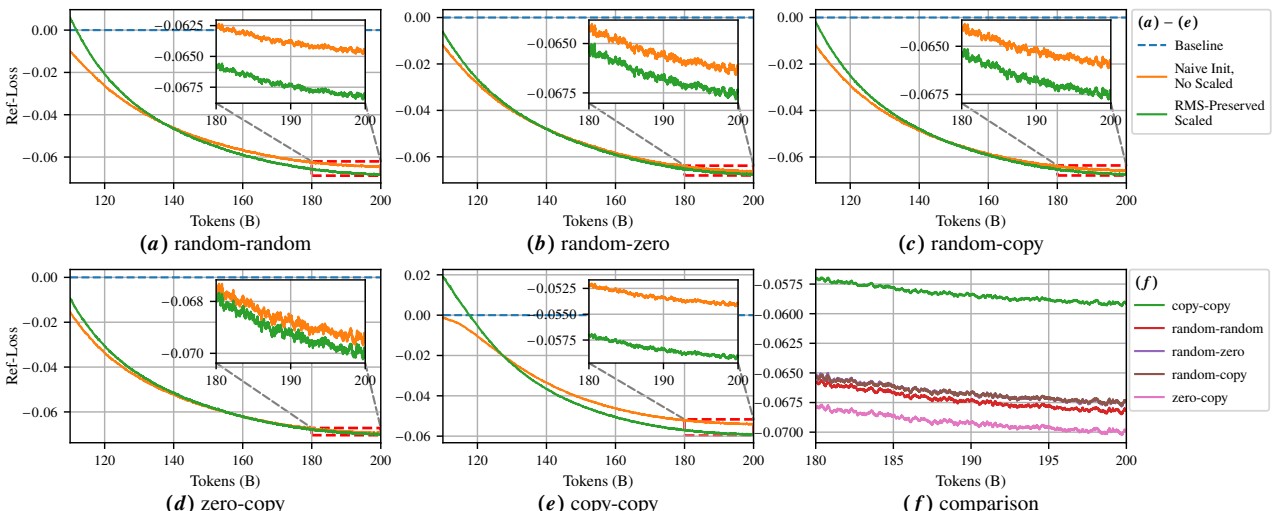

Figure 6. **Hidden-dimension expansion mirrors expert-inner growth.** We repeat the RMS-preserving scaling comparison under hidden-dimension $2\times$ expansion ($1024 \rightarrow 2048$ at $100\,\mathrm{B}$ tokens). Across initialization pairs, RMS-preserving rescaling consistently improves late-stage convergence relative to naive unscaled expansion, exhibiting the same pattern observed for expert-inner growth in Sec. 3.3.

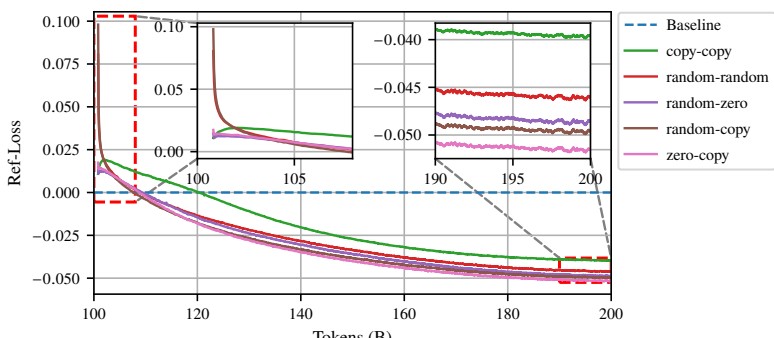

Figure 7. **Relationship between immediate perturbation and final loss.** Under expert-inner expansion, we zoom into the immediate post-expansion perturbation and final loss under different initialization pairs. The magnitude of the immediate loss spike has no direct relationship with the final loss. For example, random-copy produces a higher initial perturbation but achieves a lower final loss than random-zero.

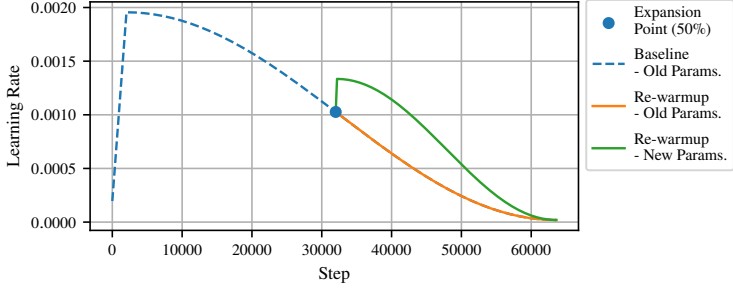

Figure 8. A sample asymmetric learning rate re-warmup curve. At the expansion step $t_e$, the learning rate of the original parameters remains on the baseline cosine schedule, whereas the newly introduced parameters are re-warmed from the instantaneous rate $\eta_e = \eta(t_e)$ to a higher peak $\hat{\eta}_{\max} = \rho\,\eta_e$ over $\tau_w$ steps, and then decay with the same cosine tail toward $\eta_{\min}$, as specified in Eq. (28).

## G. A Sample Asymmetric Re-warmup Learning Rate Curve

To make the asymmetric re-warmup schedule in Eq. (28) more tangible, we plot a representative learning rate trajectory in Fig. 8. Typically, at the expansion point, the original parameters retain the unchanged baseline cosine schedule for continuity, while the newly introduced parameters are re-warmed up from the current learning rate to a slightly higher peak for a short window, followed by decay.

## H. Hyperparameter Search for Asymmetric Re-warmup

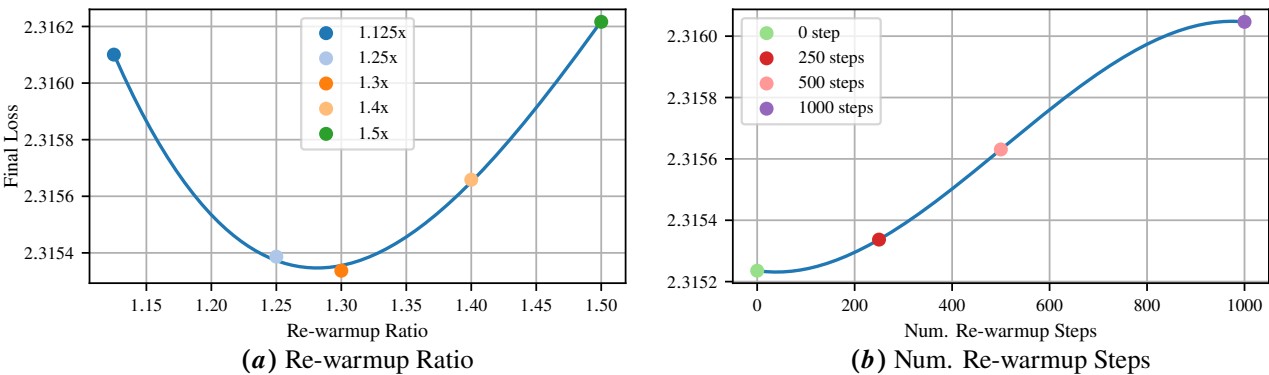

**(a)** Re-warmup Ratio    **(b)** Num. Re-warmup Steps

*Figure 9.* **Hyperparameter search for asymmetric re-warmup.** Under the expert-inner $2\times$ expansion setting, $\rho \approx 1.25$–$1.3$, $\tau_w \approx 0$–$250$, yields the lowest final loss and we adopt $\rho = 1.3$, $\tau_w = 250$ as the default re-warmup configuration in all experiments involving re-warmup.

We study how the asymmetric re-warmup schedule in Eq. (28) depends on two hyperparameters: the re-warmup ratio $\rho$ and the number of re-warmup steps $\tau_w$. We conduct this search under the expert-inner $2\times$ expansion setting.

Fig. 9 summarizes the final loss obtained by sweeping $\rho$ and $\tau_w$. The results exhibit a broad, stable region in which re-warmup is most effective: $\rho \approx 1.25$–$1.3$ and $\tau_w \approx 0$–$250$ steps achieve the lowest final loss, indicating that newly introduced parameters benefit from a modest, short-lived learning rate boost rather than a prolonged or overly strong re-warmup. Empirically, we find that this setting is also suitable for hidden-dimension expansion, and we therefore set $\rho = 1.3$ and $\tau_w = 250$ as the default hyperparameters for all experiments involving re-warmup.

## I. Ablation for RMS-Preserving Scaling and Asymmetric Strategies

*Table 6.* **Isolating the effects of RMS-preserving scaling and asymmetric strategies.** We compare results under $2\times$ expert-inner expansion across four settings to isolate the contribution of each component: (i) *W/O RMS&Asym.*, naive expansion with neither component applied, (ii) *W/ RMS*, RMS-preserved scaling only, (iii) *W/ Asym.*, asymmetric optimizer state reset and learning rate re-warmup only, and (iv) our *SPARKLING* framework. While asymmetric strategies narrow the gap of whether RMS-preserving scaling is applied, combining both approaches yields *additive* benefits, achieving the lowest final loss and best downstream performance across all initialization pairs.

| Model | Loss (↓) | ARC-C (↑) | ARC-E (↑) | Arith. (↑) | BoolQ (↑) | CSQA (↑) | HellaS. (↑) | MMLU (↑) | OBQA (↑) | PIQA (↑) | SciQ (↑) | SIQA (↑) | WinoG. (↑) | Avg. (↑) |
|---|---|---|---|---|---|---|---|---|---|---|---|---|---|---|
| Baseline (small) | 2.3673 | 41.47 | 72.46 | 43.63 | 66.36 | 47.58 | 65.86 | 32.75 | 39.40 | 76.28 | 92.50 | 46.57 | 62.19 | 57.26 |
| Baseline (expand) | **2.3096** | 43.14 | 74.56 | 55.67 | **67.80** | 47.58 | **69.45** | 32.67 | **42.40** | 78.18 | 92.80 | 46.67 | 64.80 | 59.64 |
| *random-copy* | | | | | | | | | | | | | | |
| W/O RMS&Asym. | 2.3185 | 43.81 | 74.21 | 53.47 | 66.06 | 47.75 | 68.34 | 33.33 | 40.40 | 78.13 | 93.10 | 47.03 | 63.30 | 59.08 |
| W/ RMS | 2.3177 | 42.14 | 71.93 | 58.33 | 65.14 | 47.67 | 68.78 | 32.35 | 40.20 | 77.58 | 92.30 | 45.65 | **65.59** | 58.97 |
| W/ Asym. | 2.3174 | 43.14 | 74.04 | 51.07 | 66.67 | 48.32 | 68.47 | 33.47 | 41.00 | 76.93 | 93.40 | 47.29 | 63.38 | 58.93 |
| SPARKLING | 2.3171 | 42.81 | 74.56 | 52.67 | 66.64 | **49.71** | 68.82 | 33.33 | 40.20 | **78.51** | 93.20 | 47.03 | 64.40 | 59.32 |
| *zero-copy* | | | | | | | | | | | | | | |
| W/O RMS&Asym. | 2.3165 | 41.81 | 74.56 | 47.77 | 64.83 | 48.57 | 68.74 | 32.84 | 38.60 | 76.93 | 92.80 | 47.34 | 64.01 | 58.23 |
| W/ RMS | 2.3157 | 43.81 | **75.26** | 53.00 | 66.73 | 48.98 | 68.59 | 32.98 | 41.20 | 77.31 | 93.20 | 46.88 | 64.17 | 59.34 |
| W/ Asym. | 2.3163 | 43.48 | 74.74 | 60.20 | 66.06 | 47.99 | 68.74 | 33.69 | 41.00 | 77.48 | 92.80 | 47.75 | 64.25 | 59.85 |
| SPARKLING | 2.3154 | 43.48 | **75.26** | 58.20 | 66.51 | 48.08 | 68.79 | **34.31** | 40.80 | 77.75 | **93.60** | 47.19 | 64.64 | 59.88 |
| *copy-copy* | | | | | | | | | | | | | | |
| W/O RMS&Asym. | 2.3318 | **44.48** | 74.56 | 50.10 | 67.16 | 48.48 | 68.01 | 32.98 | 40.80 | 77.42 | 92.90 | 46.62 | 64.09 | 58.97 |
| W/ RMS | 2.3276 | 44.15 | 73.86 | 51.10 | 66.45 | 48.24 | 68.04 | 32.71 | 41.80 | 77.09 | 92.90 | 46.98 | 64.25 | 58.96 |
| W/ Asym. | 2.3166 | 43.81 | 74.04 | 56.93 | 67.09 | 48.48 | 69.03 | **34.58** | 41.80 | 77.37 | 92.90 | 46.88 | 64.40 | 59.78 |
| SPARKLING | 2.3153 | 43.48 | 74.39 | **60.50** | 66.82 | 49.06 | 69.21 | 34.05 | 41.60 | 78.35 | 93.30 | **47.80** | 65.04 | **60.30** |

**Experimental setup.** To isolate the contributions of the two principles in SPARKLING, i.e., *signal preservation* via RMS-preserving scaling and *symmetry breaking* via asymmetric optimizer state reset together with asymmetric learning rate re-warmup, we compare four configurations under each fan-out/fan-in initialization pair: (i) *W/O RMS&Asym.*, (ii) *W/ RMS*, (iii) *W/ Asym.*, and (iv) the full *SPARKLING* framework (i.e. combining both strategies). All other training-recipe details follow Sec. 4.3.

**Results.** Table 6 shows the two principles deliver complementary, *additive* gains. *W/ Asym.* substantially compresses the gap between variants with and without RMS-preserving scaling, but does not eliminate it, while *W/ RMS* still yields a loss reduction on top of the asymmetric strategies, a meaningful margin in the LLM pre-training regime. *SPARKLING* attains the lowest final loss and the best downstream average across all initialization pairs, confirming that signal preservation and symmetry breaking are orthogonal axes whose combination is strictly stronger than either alone.

## J. Comparison to Prior Function-Preserving Symmetry-Breaking Heuristics

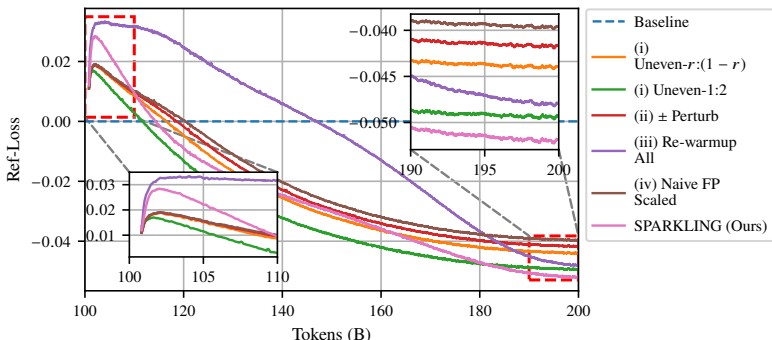

*Figure 10.* **Comparison with alternative symmetry-breaking strategies.** Under expert-inner copy-copy expansion, we compare four alternatives against our SPARKLING framework: (i) Uneven Splitting (fixed $1:2$ or randomized $r:(1-r)$ with $r \in [0.1, 0.5]$), (ii) symmetric $\pm$ perturbation that cancels in the forward pass, (iii) globally re-warmup all parameters, and (iv) naive function-preserving scaled initialization with no symmetry-breaking intervention. All alternatives converge to a higher final loss, underperforming SPARKLING. Insets highlight the post-expansion dynamics: our method exhibits a brief loss increase followed by rapid recovery, consistent with more effective symmetry breaking in the newly added capacity.

**Experimental setup.** Prior width-expansion methods that rely on copy-based expansion attempt to break symmetry by two widely used function-preserving heuristics: (i) *Uneven Splitting* (Chen et al., 2016; 2022; Du et al., 2024; Wang et al., 2024) by assigning different scaling factors to the channel being copied and the copied one, (ii) *Perturb* (Wu et al., 2020; Yuan et al., 2023; Wu et al., 2019) by adding symmetric perturbations of equal magnitude and opposite sign to the two duplicated halves. Following the expert-inner expansion in Sec. 4.3, we implement these strategies as well as (iii) *Re-warmup All*, which applies the same re-warmup schedule to all parameters, and (iv) *Naive Function-Preserving Scaled*, which applies no symmetry-breaking intervention.

**Results.** Fig. 10 shows that these heuristics, despite introducing asymmetry by construction, remain consistently weaker than our framework. Moreover, the zoomed-in view around the expansion moment highlights a transient loss up-shift followed by fast recovery, consistent with targeted exploration for newly introduced parameters benefiting from asymmetric re-warmup.

## K. Comparison to Prior Dynamics-Based Strategies

**Experimental setup.** We complement the function-preserving initialization heuristics of Appendix J by comparing SPARKLING against three representative dynamics-based strategies that intervene primarily in post-expansion optimization. Following the expert-inner expansion setup of Sec. 4.3, we implement (i) *Scaled Optimizer States + Remapped LR* (Shen et al., 2022) by scaling the optimizer states upon expansion and remapping the LR schedule to the loss-matched point on the target-model trajectory; (ii) *Uneven Splitting + Faster LR Decay* (Wang et al., 2024) by combining uneven-split initialization with the same peak learning rate but a $25\%$–$50\%$ accelerated decay schedule; and (iii) *FP-Random Initialization + Norm-Adapted LR* (Yuan et al., 2023) by combining function-preserving random initialization with a stage-wise learning rate adaptation scaled by weight norm. All baselines are evaluated under the same training recipe and token budget as our

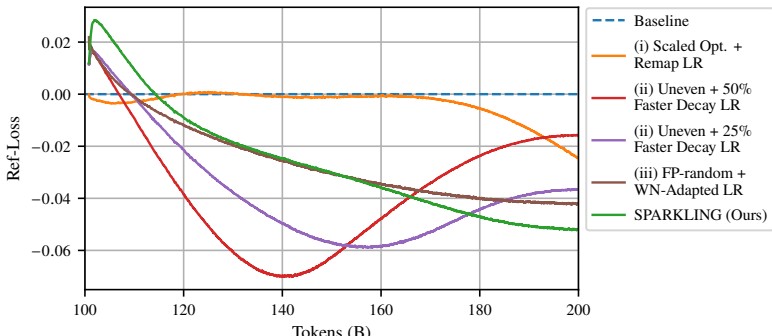

*Figure 11.* **Comparison with alternative dynamics-based strategies.** Under expert-inner expansion, we compare three alternatives against our framework: (i) optimizer state scaling, combined with remapping the LR schedule to the loss-matched point on the target model trajectory (Shen et al., 2022), (ii) uneven-split initialization with the same peak learning rate and faster decay schedule (Wang et al., 2024), and (iii) function-preserving random initialization with learning rate adapted by weight norm (Yuan et al., 2023). Our SPARKLING framework consistently outperforms all these baselines, achieving the lowest final loss.

framework.

**Results.** Fig. 11 shows that SPARKLING attains the lowest final loss against all three baselines. Each alternative addresses only one slice of the post-expansion dynamics, whereas our framework jointly enforces RMS-scale consistency and targeted backward symmetry breaking, the two complementary principles for stable mid-stage expansion.

## L. Downstream Performance of Initialization and Dynamic-based Strategies

*Table 7.* **Comparison of initialization and dynamics-based strategies.** Under $2\times$ expert-inner expansion, we compare several initialization and dynamics-based strategies. Across all strategies, our SPARKLING framework achieves the lowest final loss and outperforms all baselines on downstream tasks.

| Model | Loss (↓) | ARC-C (↑) | ARC-E (↑) | Arith. (↑) | BoolQ (↑) | CSQA (↑) | HellaS. (↑) | MMLU (↑) | OBQA (↑) | PIQA (↑) | SciQ (↑) | SIQA (↑) | WinoG. (↑) | Avg. (↑) |
|---|---|---|---|---|---|---|---|---|---|---|---|---|---|---|
| Baseline (small) | 2.3673 | 41.47 | 72.46 | 43.63 | 66.36 | 47.58 | 65.86 | 32.75 | 39.40 | 76.28 | 92.50 | 46.57 | 62.19 | 57.26 |
| Baseline (expand) | **2.3096** | 43.14 | 74.56 | 55.67 | **67.80** | 47.58 | **69.45** | 32.67 | **42.40** | 78.18 | 92.80 | 46.67 | 64.80 | 59.64 |
| *Initialization-based strategies* | | | | | | | | | | | | | | |
| Naive FP scaled | 2.3276 | 44.15 | 73.86 | 51.10 | 66.45 | 48.24 | 68.04 | 32.71 | 41.80 | 77.09 | 92.90 | 46.98 | 64.25 | 58.96 |
| ± Perturb | 2.3256 | 42.14 | 73.86 | 51.17 | 66.57 | 48.73 | 68.51 | **34.13** | 41.00 | 77.20 | 92.90 | 46.78 | 64.17 | 58.93 |
| Uneven-$r$:$(1-r)$ | 2.3234 | 42.47 | 74.04 | 42.70 | 66.02 | 50.12 | 68.25 | 33.82 | 42.00 | 77.31 | 92.90 | 46.62 | 63.77 | 58.34 |
| Uneven-1:2 | 2.3180 | 41.81 | **75.26** | 57.17 | 66.36 | **50.53** | 68.68 | 33.69 | 41.20 | 77.37 | 93.50 | 47.54 | 63.93 | 59.75 |
| *Dynamic-based strategies* | | | | | | | | | | | | | | |
| Scaled Opt. + Remap LR [1] | 2.3426 | 43.14 | 74.39 | 55.20 | 66.54 | 48.40 | 67.79 | 33.02 | 40.40 | 76.99 | 92.60 | 47.54 | 62.67 | 59.06 |
| Uneven + 50% Faster Decay LR [2] | 2.3516 | 43.48 | 73.33 | 47.53 | 66.64 | 47.42 | 67.07 | 31.82 | 39.40 | 76.61 | 92.80 | 46.88 | 63.06 | 58.00 |
| Uneven + 25% Faster Decay LR [2] | 2.3307 | 42.81 | 73.86 | 53.10 | 66.51 | 49.88 | 68.16 | 33.51 | 40.60 | 77.53 | 93.50 | 47.44 | 62.67 | 59.13 |
| FP-random + WN-Adapted LR [3] | 2.3253 | 42.81 | 74.56 | 58.13 | 66.09 | 48.32 | 68.12 | 32.71 | 39.60 | 77.20 | **93.70** | 47.54 | 64.25 | 59.42 |
| Re-warmup All | 2.3193 | **44.82** | 74.39 | 59.47 | 66.36 | 48.65 | 68.93 | 33.60 | 41.20 | 78.02 | 92.90 | 46.42 | 64.09 | 59.90 |
| *Ours* | | | | | | | | | | | | | | |
| SPARKLING | 2.3153 | 43.48 | 74.39 | **60.50** | 66.82 | 49.06 | 69.21 | 34.05 | 41.60 | **78.35** | 93.30 | **47.80** | **65.04** | **60.30** |

Beyond the pre-training loss comparisons in Appendix J and Appendix K, we further evaluate downstream performance of representative initialization and dynamics-based baselines under $2\times$ expert-inner expansion. All baselines share the training recipe of Sec. 4.3, and downstream performance is reported on the same evaluation tasks as Sec. 5.1. As shown in Table 7, SPARKLING attains the lowest final pre-training loss and the strongest average downstream performance across all initialization and dynamics-based baselines.

## M. Effectiveness Under Muon

**Experimental setup.** To verify that our framework is not tied to element-wise optimizers like AdamW, we repeat the expert-inner expansion experiment following Sec. 3.3 and 4.3 using Muon as the optimizer while keeping the other recipe unchanged. We evaluate two representative components of our method. First, we isolate *RMS-preserving scaling* by comparing against the naive unscaled initialization under the same initialization regime. Second, we evaluate *asymmetric learning rate re-warmup* for the newly introduced parameters by comparing runs with versus without the re-warmup schedule, while applying all other components of our framework.

**Results.** Fig. 12 shows the conclusions on Muon. In Fig. 12(*a*), RMS-preserving scaling produces a stable and consistent improvement over naive unscaled initialization, ultimately converging to a lower final loss under the same training budget. In Fig. 12(*b*), enabling asymmetric re-warmup further improves late-stage convergence over any other counterparts without re-warmup. Taken together, these results demonstrate that both RMS-preserving scaling and re-warmup remain effective under Muon, confirming that our framework applies beyond AdamW and extends to spectral-style updates like Muon without requiring optimizer-specific designs.

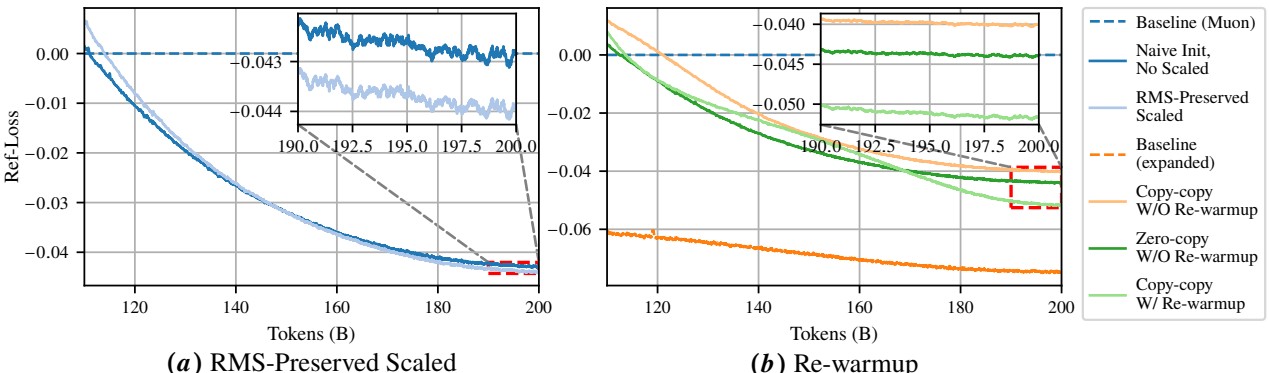

**(*a*)** RMS-Preserved Scaled      **(*b*)** Re-warmup

*Figure 12.* **Effectiveness under Muon.** We repeat the expert-inner expansion experiment (512→1024 at 100B tokens) using Muon and plot reference-loss versus training tokens. (*a*) RMS-preserving scaling consistently improves late-stage convergence compared to naive unscaled initialization. (*b*) With RMS-preserving scaling and asymmetric state reset applied, asymmetric learning rate re-warmup further lowers the final loss, confirming that our framework remains effective under Muon.

## N. Generalization to Dense Models

*Table 8.* **Loss and downstream performance under** $2\times$ **inner-dimension expansion for dense models.** SPARKLING matches or outperforms the from-scratch baseline on most tasks, demonstrating the effectiveness of our framework on dense models.

| Model | Loss (↓) | ARC-C (↑) | ARC-E (↑) | Arith. (↑) | BoolQ (↑) | CSQA (↑) | HellaS. (↑) | MMLU (↑) | OBQA (↑) | PIQA (↑) | SciQ (↑) | SIQA (↑) | WinoG. (↑) | Avg. (↑) |
|---|---|---|---|---|---|---|---|---|---|---|---|---|---|---|
| Baseline (small) | 2.5496 | 30.10 | 60.35 | 28.40 | 57.68 | 40.62 | 54.78 | 28.79 | 34.40 | 71.49 | 89.70 | 44.06 | 58.48 | 49.91 |
| *Dense, Inner* $2\times$ | | | | | | | | | | | | | | |
| Baseline (expand) | **2.4613** | 35.12 | **67.54** | **30.47** | **64.86** | 44.64 | **59.92** | 29.42 | 38.40 | 73.83 | 90.30 | 45.70 | 60.54 | 53.39 |
| Naive Expansion | 2.4809 | 34.11 | 64.39 | 28.93 | 60.12 | 42.10 | 59.28 | 28.17 | 36.60 | 73.01 | 90.60 | 45.65 | **61.09** | 52.00 |
| SPARKLING | 2.4801 | **36.79** | 66.67 | 30.40 | 64.43 | **45.29** | 59.31 | **30.93** | **38.60** | **73.99** | **91.60** | **46.57** | 60.77 | **53.78** |

**Experimental setup.** We chose MoE as the primary architecture in Sec. 3.3 and Sec. 4.3 for several reasons: it has become widely adopted in current large-scale applications, it presents greater optimization challenges than dense models and thus a more rigorous benchmark for evaluating training methodologies, and it possesses a higher capability ceiling in scaling-up scenarios, which is the regime that SPARKLING is designed to support.

Nevertheless, to assess whether the same recipe also generalizes to dense architecture, we conduct an additional $2\times$ inner-dimension expansion on a dense counterpart of our baseline with intermediate size set to 4096, while keeping all other components identical to the MoE setting in Sec. 4.3 and Appendix D.

**Results.** Table 8 reports the final pre-training loss and downstream performance on the same evaluation tasks used in Sec. 5.1. SPARKLING outperforms both the naive expansion variant and the from-scratch baseline at the same target width, despite operating under a reduced compute budget. This confirms that our framework is not tied to the MoE architecture and generalizes consistently to dense models.

## O. Generalization to Multi-Stage Expansion

**Experimental setup.** Theoretically, the benefits of iterative expansion follow naturally from induction: as long as a single expansion step yields positive gains, the same principle can be applied recursively to multiple successive expansions. To validate this hypothesis empirically, we run a 2-stage iterative expansion on top of the recipe used in Sec. 4.3: starting from an expert-inner dimension of 256, we first expand to 512 at 50 B tokens, and then expand again to 1024 at 100 B

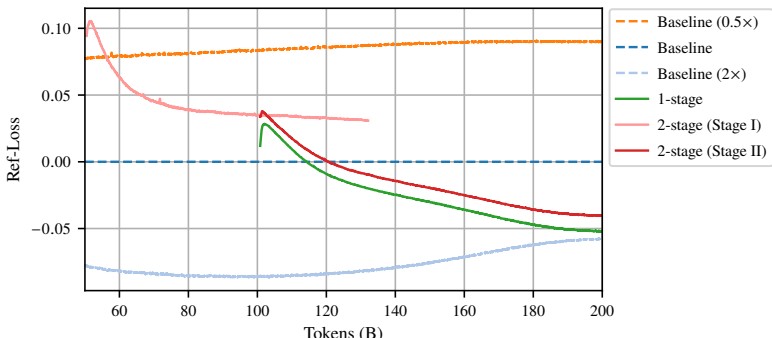

*Figure 13.* **Iterative 2-stage expansion.** We expand the expert-inner dimension from $256 \rightarrow 512$ at 50B tokens, and then $512 \rightarrow 1024$ at 100B tokens, comparing it with the direct 1-stage expansion from $512 \rightarrow 1024$ at 100B tokens. Despite a slightly higher final loss than the direct 1-stage expansion, it achieves competitive results with further reduced overall computational cost.

tokens, continuing training to a total of $200\,\mathrm{B}$ tokens. We compare this iterative trajectory to the direct 1-stage $512 \rightarrow 1024$ expansion at $100\,\mathrm{B}$ tokens already studied in Sec. 4.3, while keeping all other components of SPARKLING identical at each expansion point.

**Results.** Fig. 13 shows that our framework remains highly effective in the iterative regime: the 2-stage trajectory tracks the 1-stage curve closely and converges to only a slightly higher final loss, while operating from an even smaller initial model and thus consuming substantially less compute. This presents a natural trade-off between final pre-training loss and compute efficiency, and indicates that SPARKLING composes well across multiple successive expansions rather than being limited to a single stage recipe. We view searching for the optimal balance in this trade-off, as well as scaling to more than two stages, as a promising direction for future work.

## P. Iso-Compute Performance and Scalability

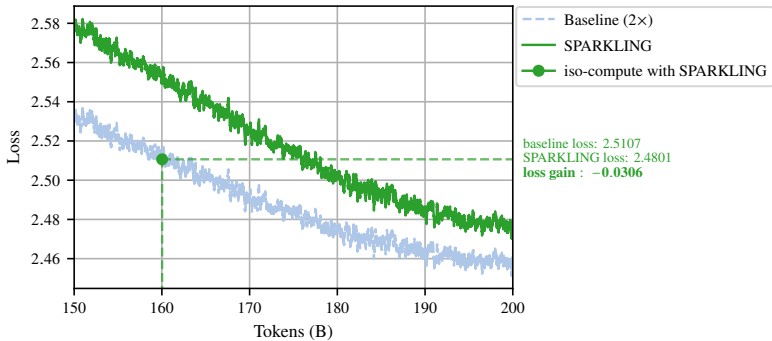

*Figure 14.* **Iso-compute comparison on dense models.** For dense models, under the *exact same* compute budget, our SPARKLING framework achieves a $0.0306$ lower loss compared to the from-scratch baseline.

**Experimental setup.** Sec. 5.4 reports the *iso-token* setting, where SPARKLING and the from-scratch baseline consume the same number of training tokens but different amounts of compute. To directly probe scalability under matched compute, we additionally provide an *iso-compute* comparison: SPARKLING and the from-scratch baseline are given the same FLOPs budget. We make this comparison across both dense and MoE architectures, as well as single and multi-stage expansion.

**Results.** For dense architecture in Fig. 14, SPARKLING achieves a $0.0306$ lower absolute loss than the from-scratch baseline at matched compute. The same trend holds on MoE across all three width axes in Fig. 15. For iterative expansion in Fig. 16, the 2-stage variant reaches a $0.0175$ lower loss than the iso-compute baseline, a strictly larger margin than the $0.0163$ improvement delivered by the 1-stage variant, while further saving compute. These iso-compute gains indicate that SPARKLING is not merely a token-efficient training strategy but yields a more favorable compute–loss scaling behavior with a larger scaling exponent $\alpha$ in $L \propto C^{-\alpha}$, and that performing more iterative expansions can produce larger loss reductions while continuing to save compute.

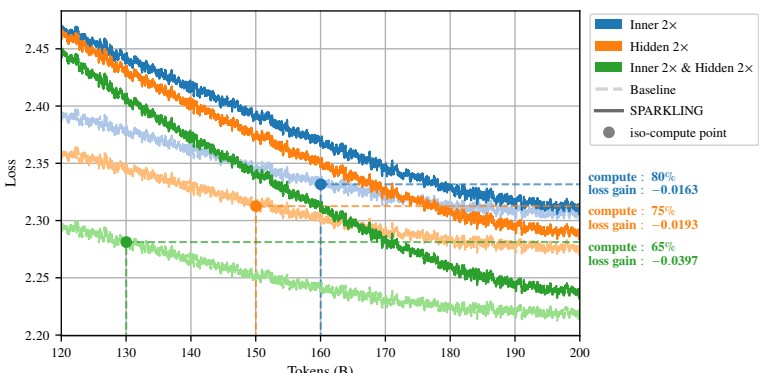

*Figure 15.* **Iso-compute comparison across mid-stage width-expansion axes.** Under the *exact same* compute budget, SPARKLING reaches a lower final loss than the from-scratch baseline across Inner 2×, Hidden 2×, and joint Hidden 2× & Inner 2× expansion, confirming that the iso-compute gains generalize across width axes on MoE.

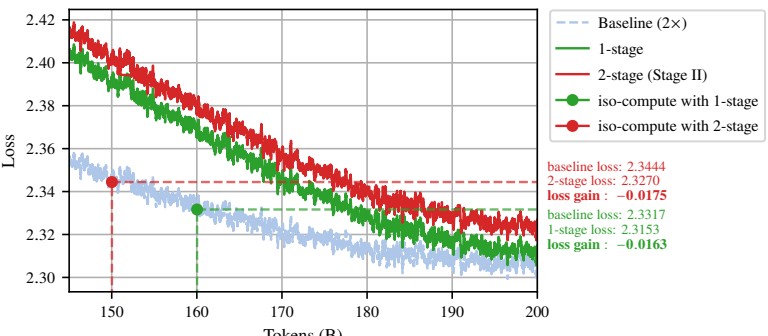

*Figure 16.* **Iso-compute comparison of iterative expansion strategies.** Under the *exact same* compute budget, both 1-stage and 2-stage expansions achieve lower loss than the from-scratch baseline. Notably, 2-stage expansion achieves a 0.0175 lower loss, providing a greater improvement than 1-stage expansion while yielding an additional 25% compute savings.

