# OpenReview forum: "SPARKLING: Balancing Signal Preservation and Symmetry Breaking for Width-Progressive Learning"
_ICML.cc/2026/Conference — ICML 2026 regular_

### Official Review · Reviewer_VXUU · 2026-03-06

**Soundness:** 3
**Presentation:** 3
**Significance:** 3
**Originality:** 3
**Overall Recommendation:** 4
**Confidence:** 3

**Summary:**

This paper addresses the problem of progressive learning (PL). The goal here is to increase the size of the model mid-training during LLM pretraining, to reuse small model weights and thus enable efficient training. While doing so, initializing the newly introduced parameters appropriately is crucial in order to ensure that the model converges and still performs at par with training the large model from scratch. The authors specifically tackle PL for width expansion, where the width of intermediate layers is expanded (usually doubled) mid-training.
They address two issues, first activation scaling and second gradient symmetry with their proposed initialization strategy which in turn helps improve convergence.

**Compliance With Llm Reviewing Policy:**

Affirmed.

**Final Justification:**

The authors have largely addressed all my questions. I believe there can be better explanations or preliminary experiments to highlight the measure if the loss spikes introduced by the proposed method are helpful. Hence, I will maintain my score.

**Key Questions For Authors:**

See above section.

**Limitations:**

yes

**Strengths And Weaknesses:**

Strengths

1. The authors propose RMS scaling for maintaining coordinate wise activation variance after width expansion, resetting optimizer states, and reintroducing an LR schedule for new params to promote better feature learning and signal propagation, while smoothly transitioning to a wider layer during training.
2. The experiments show improved convergence with their proposed method, which is abl


Weaknesses and Questions

1. In Figure 1, the copy-copy initialization seems to do much worse than the others. However, in the subsequent method, copy-copy is chosen over the others, is there a reason for this?
    - Additionally, how do these initializations have an impact on training dynamics immediately after width expansion. In most cases recovery seems possible. It would help to have a discussion about the effect of the perturbation due to these dynamics and the subsequent model convergence.
2. Is RMS preserved scaling alone necessary? As per Fig 1, even in its absence the model convergence only slightly worse. In that case, are asymmetric LR and resetting optimizer states sufficient?
3. As per Eq 4, RMS preserved scaling should help models with pre-norm layers. How does this transfer to the set of models with post-norm layers, is the preserved scaling required here. Experiments are only presented on a pre-norm model.
4. Similarly, an ablation isolating the effects of asymmetric LR and resetting optimizer states to observe their effects on the downstream performance would be instructive in highlighting their overall importance in the method.

Overall, the paper addresses a relevant problem to enable efficient pretraining. The proposed method shows improvements in performance as well as efficiency gains, however, some key ablations are missing which would help emphasize the effectiveness of the method.

---

> ### Author Rebuttal · Authors · 2026-03-31
>
> Thanks for your careful review and insightful suggestions. We will address each of your concerns as below.
>
> **1. Explanation for Copy-Copy Initialization and Training Dynamics (Q1)**
>
> We found it kind of counterintuitive that copy-copy initialization performs the worst among all initializations as in Fig.1(f), despite the most complete preservation of the originally learned distribution statistics for a seamless transition. Delving deeper, we find this performance degradation occurs due to the symmetry lock issue, as demonstrated in detail in Sec.4. However, **once combined with our subsequent asymmetric strategies, copy-copy** successfully mitigates this problem, **achieves the best results**, and benefits the most from these approaches. This explains why copy-copy is chosen over the others.
>
> **Impact of Perturbation on Dynamics and Convergence**
>
> We observed that immediately after perturbation, it is difficult to distinguish which initialization is better. Although the perturbation impacts training dynamics (e.g., causing immediate loss spikes), the models recover quickly in most cases. We claim that **the magnitude of the immediate loss spike has no direct relationship with the final loss**
>
> [*[Link] as shown in the figure here.*](https://anonymous.4open.science/r/SPARKLING/fig13_perturb_loss.png)
>
> It also indicates that function-preserving is not the key point for expansion.
>
> Instead, the **perturbation primarily affects training dynamics by bringing the model to different regions of the loss landscape**, which leads to different final convergence losses. Since our primary focus is the eventual model capability, we care more about the final convergence loss as in Fig.1(f).
>
> **2. Ablation on Isolating RMS-Preserved Scaling and Asymmetric Strategies (Q2 & Q4)**
>
> To better illustrate their individual effects and overall importance, we provide an ablation on both pre-training loss and downstream performance here, isolating the effects of RMS-preserved scaling, asymmetric resetting & LR, and their combination.
>
> [*[Link] Here are the results.*](https://anonymous.4open.science/r/SPARKLING/tab7_isolating_rms_asym.png)
>
> As shown in the results, **combining both strategies yields an *additive* benefit.**
> While the asymmetric strategies indeed compress the performance gap of whether RMS scaling is applied, the gap still persists. We still consider the **gains from RMS-preserved scaling to be orthogonal to those from the asymmetric strategies, producing a synergistic "1+1=2" improvement.**
>
> Our results demonstrate that RMS-preserved scaling is an important and general principle across different initializations. Although its relative improvement may appear smaller compared to the substantial contributions of the asymmetric strategies, **it still yields a notable improvement (e.g., a loss reduction of 0.001\~0.005) even when asymmetric strategies are applied**. In the context of LLM pre-training, such a reduction in loss is actually **a significant improvement rather than a marginal one**.
>
> **3. Applicability of RMS-Preserved Scaling to Post-Norm Models (Q3)**
>
> We appreciate the reviewer's careful consideration of different architectures. **The RMS-preserved scaling is equally applicable and necessary for post-norm models.**
>
> In a post-norm block, $\mathbf{h} \leftarrow \mathrm{Norm}(\mathbf{h} + f(\mathbf{h}))$, the core mechanism remains the addition of the residual $\mathbf{h}$ and the functional output $f(\mathbf{h})$. The relative magnitude (i.e., RMS ratio) between $\mathbf{h}$ and $f(\mathbf{h})$ controls their respective contributions. Altering the RMS of $f(\mathbf{h})$ during expansion disrupts this critical ratio *before* normalization is applied. Thus, our **RMS-preserved scaling is a general and essential strategy to consistently control the input-output RMS ratio across different architectures. The logic holds *true* regardless of whether a pre-norm or post-norm is applied.** We will clarify this broader applicability in the revision.
>
> **Explanation for Pre-Norm Experimental Setup**
>
> We primarily presented our analysis using the pre-norm formulation in the manuscript simply because it is **widely adopted in modern LLMs**, and our experimental setup **follows the original OLMoE pre-norm configuration for a fair comparison**. Specifically, switching the architecture to post-norm would introduce additional changes to the initialization, model design, or hyperparameter settings, potentially leading to training instability or collapse. To ensure a fair and stable comparison, we followed the baseline's default pre-norm setting without modifications.
>
> ---
>
> We really appreciate your thorough review of our paper and thoughtful insights. We have carefully replied to your questions and will incorporate your suggestions in the camera-ready version. We look forward to a more positive assessment of our submission based on our responses, and we are happy to provide more details regarding any further questions.

---

> > ### Author Rebuttal · Reviewer_VXUU · 2026-04-03
> >
> > Thank you for the detailed response, most of my questions have been addressed and I only have a couple of follow up questions,
> >
> > 1. Perturbation Dynamics: While, I agree the loss spike right after expansion does not alone determine how well the model will converge, is there a better way to measure if this loss spike (from changed training dynamics in the loss landscape) is better for training or not -- like measuring the sharpness of the loss landscape. And based on this, do the proposed changes align with improving the loss landscape. I understand setting up an experiment for this might be difficult, but would like to know what the authors think about this?
> >
> > 2. Another follow up, is that if function-preserving is not the goal then is there a benefit of still doing it while expanding. This part is not very clear to me.

---

> > > ### Author Response · Authors · 2026-04-03
> > >
> > > We sincerely thank the reviewer for these insightful follow-up questions. We fully agree with your intuition regarding the deeper mechanisms of model expansion, which we have also continuously considered in our work. We are very glad to reach this consensus and would like to elaborate on our detailed perspectives below:
> > >
> > > **1. On Perturbation Dynamics and Loss Landscape Sharpness**
> > >
> > > We agree that the immediate loss spike is only a coarse signal. A more informative view is to **examine how expansion alters the local optimization geometry**, e.g. the sharpness/flatness you mentioned, the RMS scale we proposed, or the recovery efficiency, etc. For example,
> > > - In Appendix C, Fig.4 (P16), we directly observed the RMS scale after expansion, and found that the **recovery of RMS scale** can indeed lead to better final convergence as RMS-preserving satisfed.
> > > - Also, as noted in prior work [*Hochreiter and Schmidhuber (1997), Flat Minima, Neural Computation 9(1): 1–42; Keskar et al. (2017), On Large-Batch Training for Deep Learning: Generalization Gap and Sharp Minima, ICLR 2017*], **flatter minima** generally correlate with better generalization. Therefore, measuring landscape sharpness is indeed a better way to judge if a perturbation is beneficial.
> > >
> > > Our current understanding is that **a “good” expansion places the model in a region that preserves meaningful pretrained structures while remaining easy to optimize**, rather than simply minimizing the instantaneous perturbation. *The loss spike itself only reflects a shift in the model's starting region for subsequent optimization.* In this context, we can categorize the loss spike into two types:
> > > 1. **Destructive Spikes**: Caused by improper initialization, which pushes the model into a worse landscape and harms recovery.
> > > 2. **Constructive Spikes**: Introduced by slight, controlled perturbations, like our RMS re-scaling, LR re-warmup, or other heuristic perturbation strategies. This "escape and explore" mechanism helps the model break out of the smaller model's confined basin, allowing the optimizer to explore the newly added capacity and eventually settle into a broader, flatter minimum.
> > >
> > > From this perspective, **our proposed changes align precisely with improving the effective optimization landscape**. By balancing activation RMS scales followed by asymmetric strategies, **our method provides a constructive spike while ensuring a smoother transition that guides the optimizer toward these flatter regions**. We agree that directly validating this through sharpness or curvature measurements is a highly valuable direction for our future work.
> > >
> > > **2. On the Benefit of Function-Preserving**
> > >
> > > Regarding function-preserving, we do *NOT* mean it is *unnecessary*. Rather, it is *not sufficient on its own* to guarantee better final convergence. Its primary benefit is **providing a stable and meaningful initialization that retains useful pretrained structures**, instead of introducing arbitrary reparameterizations.
> > >
> > > This benefit can be empirically supported in our paper:
> > > - As shown in Fig. 1, function-preserving variants (e.g., random-zero *(b)*, zero-copy *(d)*) achieve lower final losses than their non-preserving counterparts (random-random *(a)*, random-copy *(c)*), respectively.
> > > - **Under copy-copy initialization, function-preserving naturally aligns with RMS-preserving** when the copy ratio $c \ge 1$ (Eq. 16).
> > >
> > > However, whether the model converges well ultimately depends on the subsequent training dynamics (e.g., RMS scale balance and optimization behavior). This is why we build our RMS re-scaling design *on top of* the function-preserving construction. As demonstrated by our results, **RMS-preserving indeed acts as a more general and effective principle regardless of initializations**.
> > >
> > > In summary, an ideal expansion strategy strikes a delicate balance: (i) it **preserves useful pretrained structure to provide a stable starting point**, either by *more fundamentally RMS-preserving* or *function-preserving practically in some cases*, and (ii) it intentionally **introduces a slight perturbation to encourage the model to escape local basins and explore the newly available landscape for a better final minimum**, either by *initialization design* or *optimization dynamic-based strategies*.
> > >
> > > ---
> > >
> > > We sincerely appreciate your time and constructive feedback, which have significantly inspired our deeper thinking on the related issues of this work. We hope that our clarifications have thoroughly addressed your concerns and will encourage a more favorable evaluation of our work. Please feel free to reach out to us for any further questions.

---

### Official Review · Reviewer_b4kY · 2026-03-11

**Soundness:** 3
**Presentation:** 2
**Significance:** 2
**Originality:** 3
**Overall Recommendation:** 5
**Confidence:** 3

**Summary:**

The paper proposes SPARKLING, a systematic progressive learning framework during the mid-stage by width expansion. The method integrates two prior approaches, 1. Initialization and 2. optimization dynamic. For initialization, they suggest a novel perspective, signal preservation, which is maintaining the RMS-sclae of activations during expansion, contrary to the conventional function preserving perspective. For optimization dynamics, they remove the functionally redundancy that arises by copy-based initialization, through optimizer resetting and learning rate re-warmup. The method is validated by successive pretraining and fine-tuning on Mixture-of-Experts.

**Compliance With Llm Reviewing Policy:**

Affirmed.

**Final Justification:**

The authors have adequately addressed my concerns in their final response. I am willing to increase my score if the revised version includes additional experimental results.

**Key Questions For Authors:**

Q1. Is SPARKLING orthogonal to the existing optimization dynamic techniques (such as papers in the related works)? If they are not orthogonal, please provide a comparison with these related works as baselines in Section 5.

Q2. Why did you compare ‘Asym. Reset’ with ‘Asym. Reset + Scaled Opt.’ in Figure 2? How did the optimization state scaling come up? If I'm not missing any points in the paper, there was no explicit detail about optimization state scaling, thereby it was difficult to follow the motivation behind this comparison and the resulting conclusions. In addition, please specify the implementation details for the optimizer-state scaling.

Q3. In Table 1, SPARKLING shows a loss between the ‘Baseline (expand)’ and ‘Naive Expansion’ during pre-training, which already validate its effectiveness over prior initialization methods. But I wonder how SPARKLING outperforms the ‘Baseline (expand)’  in most of the tasks. Could you provide any justification for how this happened?

Q4. Could you explain how the ‘RMSNORM WEIGHT EXPANSION’ described in Section 3.2.4 differs from standard fan-in/fan-out expansion? As far as I understood, the experiments such as Figure 1 reported fan-out/fan-in initialization pairs, but I couldn't cache weight expansion. How was it applied to the experiment?

**Limitations:**

yes

**Strengths And Weaknesses:**

Strength
- The proposed approach, signal preservation, is supported by the  well-established role of RMS norm in activations.
- The paper provides a rigorous analysis of the limitations inherent in copy-based initialization and the reason why (even) modern optimizers can’t mitigate the limitation. The two proposed techniques for symmetry breaking are easy to implement, and directly address the identified issues.
- The experiments conducted in the paper lie on a large model scale and token number, spanning from pre-training to fine-tuning.

Weakness
- The empirical validation is exclusively focused on MoE architectures. The manuscript lacks a clear justification for why dense Transformers or other architectures were excluded.
- While SPARKLING integrates both initialization and optimization dynamics, in Section 5, the comparison is restricted to FP initialization. Unless SPARKLING is orthogonal to the existing optimization dynamics techniques, it needs to compare against other dynamic-based optimization strategies.

---

> ### Author Rebuttal · Authors · 2026-03-31
>
> Thanks for your careful review and insightful suggestions. We will address each of your concerns as below.
>
> **1. Generalization to Dense Models**
>
> As a similar question was raised by another reviewer, please kindly refer to our response to **Reviewer Kiwk (Point 1)** above for detailed experimental results and discussions demonstrating the generalization of our SPARKLING to dense models.
>
> **2. Comparison with Other Dynamic-based Techniques**
>
> We appreciate your careful reading and constructive feedback. We would like to explain that the **primary goal of Sec.5 is to demonstrate SPARKLING's *overall* effectiveness against training from scratch. Detailed ablations on initialization and optimization dynamics are provided in Sec.3-4.** Regarding your concerns about optimization dynamics, we provide further details in the following two aspects:
>
> **2.1 Implementation Details and Motivation of *Scaled Opt.* (Q2)**
>
> The "Scaled Opt." actually refers to a baseline in prior work [1]. We briefly discussed its motivation and mechanics in Sec.4.2.1 (L295-297, L310-312, P6). Specifically, it applies the **exact same scaling operations to the optimizer states as those applied to the parameters** during expansion, intuitively ensuring a seamless and consistent transition in optimization dynamics. However, our results (Fig.2) show this explicit state scaling is unnecessary. Optimization dynamics quickly adapt to the new architecture and become gradient-dominated. We will clarify these details in the revision.
>
> **2.2 Comparison with More Dynamic-based Baselines (Q1)**
>
> We acknowledge that the manuscript could benefit from a broader comparison with existing dynamic-based techniques. To address this, we have supplemented experiments comparing SPARKLING against several dynamic-based strategies:
>
>
> [1] Shen et al. (2022), Staged Training ..., ICML 2022.
>
> - The full setting includes the aforementioned optimizer state scaling combined with LR schedule remapping to the loss-matched point on the target model trajectory.
>
> [2] Wang et al. (2024), LEMON: Lossless Model Expansion, ICLR 2024.
>
> - Uneven-split initialization with a faster LR decay schedule.
>
> [3] Yuan et al. (2023), Accelerated Training via Incrementally ..., NeurIPS 2023.
>
> - Function-preserving random initialization with weight-norm-based LR adaptation.
>
> [*[Link] Here are the comparison results.*](https://anonymous.4open.science/r/SPARKLING/fig11_dynamic.png)
>
> As shown, **SPARKLING consistently outperforms all baselines**, achieving the lowest final loss. These comparisons will be incorporated into the revised manuscript to further illustrate the effectiveness of our method.
>
> **3. Justification for Mismatch in Loss and Downstream Performance (Q3)**
>
> We appreciate this insightful question. This is indeed a very interesting phenomenon that we frequently observe in our experiments: the pre-training loss and downstream performance do not always align.
>
> We hypothesize this **mismatch arises from different optimization trajectories and learning manifolds under different training recipes. Models initialized via expansion converge to different regions of the loss landscape** compared to those trained from scratch. Consequently, they can exhibit distinct downstream generalization despite slightly higher pre-training losses, suggesting progressive learning may inherently enhance generalizability. Thus, relying solely on pre-training loss for comparison is insufficient.
>
> Investigating how these training paradigms affect generalization remains an important direction for future work.
>
> **4. Explanation about RMSNorm Weight Expansion (Q4)**
>
> We discussed RMSNorm expansion separately in Sec.3.2 due to its **mathematical difference from linear layers: it involves 1D element-wise multiplication rather than 2D matrix multiplication**.
>
> In our experiments, RMSNorm expansion only occurs during hidden dimension expansion in Appendix D, Fig.5 (P16-17), not expert-inner expansion in Fig.1. For these cases, we **uniformly applied the *copy* strategy** based on a preliminary ablation study.
>
> [*[Link] Here are the ablation results.*](https://anonymous.4open.science/r/SPARKLING/fig12_rmsnorm.png)
>
> As shown, ***random* and *copy* strategies yield almost identical final losses**. Because RMSNorm contains significantly fewer parameters than linear layers, its initialization has a *minimal impact* on overall performance. We unified the use of the *copy* strategy and omitted this minor ablation to keep the main text focused. We apologize for any potential confusion caused by omitting this detail and will explicitly include these details in the revision.
>
> ---
>
> We really appreciate your thorough review of our paper and thoughtful insights. We have carefully replied to your questions and will incorporate your suggestions in the camera-ready version. We look forward to a more positive assessment of our submission based on our responses, and we are happy to provide more details regarding any further questions.

---

> > ### Author Rebuttal · Reviewer_b4kY · 2026-04-03
> >
> > Thank you for your detailed response. It addressed most of my concerns; however, I have one remaining question regarding the experiments in Sec.5.
> >
> > While you said the primary goal of Sec.5 is to demonstrate SPARKLING’s overall effectiveness against training from scratch, I  believe Table 1 should include the experiments found in the anonymous link (Fig 11) as baselines for the following reasons:
> > 1. The method is not orthogonal to the dynamic-based strategies. Therefore, to demonstrate SPARKLING’s effectiveness, it should be compared not only with naive expansion, but also with other prior strategies.
> > 2. Since the claim of SPARKLING’s superiority over training from scratch relies on downstream task results, it is important to demonstrate that this advantage holds when compared to other existing methods as well.
> > 3. While the additional experiments in the anonymous link show a superior final loss for SPARKLING, a lower loss does not necessarily guarantee better performance on downstream tasks (as observed in our previous discussion and the table).
> >
> > I respect the efforts and the solid research presented in this work. If these comparative downstream results are included in the paper and my concerns regarding the baseline comparisons are addressed, I would be happy to re-evaluate the paper and consider increasing my score.

---

> > > ### Author Response · Authors · 2026-04-03
> > >
> > > Thank you for acknowledging our efforts and for your constructive feedback.
> > >
> > > Considering your concern regarding the downstream performance of existing methods, we have supplemented the downstream task comparisons over the initialization and dynamic-based strategies mentioned in Sec.5 and our discussions above.
> > >
> > > [*[Link] Here are the downstream comparison results.*](https://anonymous.4open.science/r/SPARKLING/tab8_downstream_all.png)
> > >
> > > As shown, across all strategies, our SPARKLING framework **achieves the lowest final loss** and **consistently outperforms all baselines on overall downstream tasks**. We will include these comprehensive comparisons in the revised manuscript.
> > >
> > > Thank you again for your time and valuable suggestions, which have greatly helped strengthen our paper. We hope these demonstrations have fully resolved your concerns and help you make a more positive assessment of our work. Please feel free to reach out to us for any further questions.

---

### Official Review · Reviewer_Kiwk · 2026-03-12

**Soundness:** 3
**Presentation:** 3
**Significance:** 3
**Originality:** 2
**Overall Recommendation:** 4
**Confidence:** 4

**Summary:**

Authors identify two causes of width expansion failure -
1. Gradient Symmetry
2. Irregular Activation stats

They introduce optimizer stats resetting/re-warmups to break the symmetry and RMS to control the activations. This reduces the training compute compared to training from scratch substantially.

**Compliance With Llm Reviewing Policy:**

Affirmed.

**Final Justification:**

The authors have provided extensive experiments in IsoFLOP setting and have addressed my comments over scalibility.

**Key Questions For Authors:**

1. The proposed algorithm seems to be an iterative process of width expansion, did the authors perform iterative expansion?
2. Authors demonstrate that fan-in coordinates are created by copying, it underperforms compared to other RMS-preserved variants. However, the paper does not provide explaination for the same.
3. The experiments on MoE (0.5B) scale, how SPARKLING behaves in a purely dense architecture remains an open question.

**Strengths And Weaknesses:**

1. Formal explaination of width failure case and and highlight that advanced optimizers with orthogonalization such as Muon fail to spontaneously break this structural symmetry.
2. Although asymmetric re-warmup schedule is sensitive by construction but it is able to protect trained weights while allowig the new width for learning.
3. ~30% reduction in training time on 0.5B MoE model scale.

Weakness
1. The performance and applicability on dense architectures needs to be visited in the paper. Currently all experiments are shown on MoE model at 0.5B scale.
2. The proposed method can benefit from iterative refinements rather than 1-shot expansion, however the authors do not comment and experiment around this setting.

---

> ### Author Rebuttal · Authors · 2026-03-31
>
> Thanks for your careful review and insightful suggestions. We will address each of your concerns as below.
>
> **1. Generalization to Dense Models (Q3)**
>
> We sincerely thank the reviewer for this constructive suggestion. In our main experiments, we chose the MoE architecture as our primary focus for **several reasons**. First, MoE has become a **highly popular and widely adopted model architecture in current large-scale applications** (e.g., DeepSeek-V3/R1, Kimi-K2/K2.5, etc.). Second, training MoE models presents greater optimization challenges compared to dense models, making it **a more rigorous and challenging benchmark for evaluating training methodologies**. Finally, the core motivation of our SPARKLING framework is to facilitate **model scaling**, and MoE inherently possesses **a higher capability ceiling in scaling-up scenarios**. Therefore, MoE was selected as our preferred architecture.
>
> Nevertheless, we fully agree with your suggestion that investigating the applicability of our method on dense architectures provides valuable insights, and we are more than happy to supplement the results on dense models. To address this concern, we conducted **additional experiments on dense architectures** for $2\times$ inner dimension expansion. The table below presents the results.
>
> |Model|Loss|ARC-C|ARC-E|Arith.|BoolQ|CSQA|HellaS.|MMLU|OBQA|PIQA|SciQ|SIQA|WinoG.|Avg.|
> |:-|:-|:-|:-|:-|:-|:-|:-|:-|:-|:-|:-|:-|:-|:-|
> |Baseline(small)|2.5496|30.10|60.35|28.40|57.68|40.62|54.78|28.79|34.40|71.49|89.70|44.06|58.48|49.91|
> |*Dense, Inner$2\times$*|||||||||||||||
> |Baseline(expand)|**2.4613**|35.12|**67.54**|**30.47**|**64.86**|44.64|**59.92**|29.42|38.40|73.83|90.30|45.70|60.54|53.39|
> |Naive Expansion|2.4809|34.11|64.39|28.93|60.12|42.10|59.28|28.17|36.60|73.01|90.60|45.65|**61.09**|52.00|
> |SPARKLING|2.4801|**36.79**|66.67|30.40|64.43|**45.29**|59.31|**30.93**|**38.60**|**73.99**|**91.60**|**46.57**|60.77|**53.78**|
>
>
> The results demonstrate that our SPARKLING framework are **still highly effective for dense models**. It enables the model expanded by our framework to consistently outperforms the model trained from scratch, effectively validating the generalization and applicability of SPARKLING on purely dense architectures.
>
> **2. Investigation on Iterative Expansion (Q1)**
>
> We appreciate the reviewer's insightful comment. Theoretically, the benefits of iterative expansion follow naturally from the principles of induction. As long as a single expansion yields positive gains, these benefits can be extrapolated to multiple successive expansions.
>
> To empirically validate this hypothesis and provide a more comprehensive analysis, we further evaluated our framework under an iterative expansion setting.
>
> [*[Link] The figure here shows the results.*](https://anonymous.4open.science/r/SPARKLING/fig10_2stage.png)
>
> The results demonstrate that **our framework remains highly effective during iterative multi-stage expansion**. Although the final loss is slightly higher compared to the direct 1-stage expansion, it still maintains competitive results while further reducing the overall computational cost. This presents a natural tradeoff between final model capability and training efficiency, and we view finding the optimal balance in this tradeoff as a promising direction for future work.
>
> **3. Explanation for Copy-Based Fan-In Underperformance (Q2)**
>
> We would like to clarify that the underlying reasons for the underperformance of copy-based initialization **have actually been discussed in our submission**, specifically at **the beginning of Sec. 4 & Sec. 4.1 (L267-278, P5) and Appendix A.3 (L739-747, P14)**.
>
> To briefly summarize the logic in our manuscript again: the underperformance of pure copy-based expansion is not because the strategy itself is flawed, but rather **due to the "symmetry lock" problem during the backward pass**. Because the duplicated components receive identical gradients, they evolve identically and fail to diversify, rendering the expanded capacity functionally redundant. Sec.4 is dedicated entirely to addressing this exact problem. Once we apply our proposed asymmetric operations to break this symmetry lock, the copy-based initialization successfully mitigates the redundancy problem and achieves the best performance.
>
> It seems that our current presentation might not have been clear enough, leading to the overlooked explanation. We sincerely apologize for any potential inconvenience or confusion this may have caused, and we will carefully refine our writing in the camera-ready version to make this explanation much clearer.
>
> ***
>
> We really appreciate your thorough review of our paper and thoughtful insights. We have carefully replied to your questions and will incorporate your suggestions in the camera-ready version. We look forward to a more positive assessment of our submission based on our responses, and we are happy to provide more details regarding any further questions.

---

> > ### Author Rebuttal · Reviewer_Kiwk · 2026-04-02
> >
> > The authors have sufficiently addressed my concerns regarding dense architecture and iterative refinement. However the dense results are similar to baseline and iterative refinement not showing improvements raises questions on the scalability of this work. Hence I keep my score.

---

> > > ### Author Response · Authors · 2026-04-03
> > >
> > > We sincerely thank the reviewer for acknowledging our efforts. Regarding your remaining concerns about the dense architecture results and the scalability of iterative expansion, we would like to clarify as follows:
> > >
> > > **1. Performance Expectations under Reduced Compute**
> > >
> > > The primary goal of our framework is to **achieve efficient training with a comparable performance**. According to scaling laws [*Kaplan et al. (2020), Scaling Laws for Neural Language Models*], training loss $L$ follows a power law in compute $C$ (FLOPs), i.e., $L \propto C^{-\alpha} (\alpha > 0)$. Since our expanded models are trained under a significantly reduced compute budget (e.g. saving 20% for 1-stage, 25% for 2-stage), expecting them to substantially outperform a train-from-scratch baseline in absolute loss is naturally **unrealistic and counterintuitive** with the scaling laws.
> > >
> > > However, the fact that **our method (both dense and MoE architectures) achieves comparable loss and outperforms the baseline on overall downstream tasks while saving substantial 20-35% compute**, has already strongly *validated its effectiveness*. For iterative expansions, the compute is further reduced from 20%-saving to 25%-saving compared to the single-stage setting, naturally leading to an inevitable higher loss according to the scaling law. Besides, Reviewer b4kY also reckons that "a loss between the ‘Baseline (expand)’ and ‘Naive Expansion’ during pre-training already validates its effectiveness over prior methods" (from Q3 of the review).
> > >
> > > Therefore, we would like to claim that these results demonstrate effective improvements rather than a lack of scalability.
> > >
> > > **2. Iso-Compute Performance and Scalability**
> > >
> > > To directly address the scalability concern, we compared our framework with the from-scratch baseline **under the exact same compute budget** for a fair evaluation.
> > >
> > > [*[Link] The figure here shows the iso-compute comparison.*](https://anonymous.4open.science/r/SPARKLING/fig14-15_iso-compute-loss.png)
> > >
> > > The results clearly demonstrate our superiority **under equivalent computational constraints**:
> > >
> > > - For the dense architecture, our method achieves a $0.0306$ lower loss than the iso-compute from-scratch baseline. This indicates our SPARKLING framework yields a more favorable scaling behavior (larger $\alpha$ in the scaling law), meaning **a lower loss can be reached under an identical compute budget**.
> > > - For the 2-stage iterative expansion, our method achieves a $0.0175$ lower loss compared to its iso-compute baseline. Notably, this reduction is greater than the $0.0163$ improvement seen in the 1-stage expansion. This demonstrates that **conducting more times of expansions could bring larger loss reductions** over the from-scratch baseline while saving even more compute, serving as direct evidence for the scalability and iterative potential of our framework.
> > >
> > > In summary, our method exhibits **much better scaling performance** and **computation efficiency** compared to training from scratch. We hope this iso-compute perspective above could further clarify that our framework is **NOT** limited in scalability, and helps address your remaining concerns for making a more proper decision on our work. Should you have any further questions, please feel free to reach out to us.

---

### Decision · Program_Chairs · 2026-04-30

**Decision:**

Accept (regular)

**Comment:**

The paper addresses an important and a bit underexplored problem of model expansion or prgressive learning. The paper does a good job of identifying reasons for potential failures and offering mitigation techniques with good empirical gains. The discussion phase was productive and swung in favor of the paper: Reviewer Kiwk initially raised concerns about dense-model applicability and iterative expansion, but after rebuttal stated that the concerns were resolved. Reviewer b4kY was the most positive, viewing the work as technically solid, and after rebuttal indicated the concerns were largely addressed. Reviewer VXUU also remained positive, with requests for better explanations and ablations, finally saying all the concerns were mostly addressed. The rebuttal experiments have made the paper significantly stronger, so I would recommend the authors include those in the final version.